



# Improved Mean Field Estimates of GEMS AOD L3 Product: Using Spatio-temporal Variability

Sooyon Kim[1], Yeseul Cho[3], Hanjeong Ki[1], Seyoung Park[1], Dagun Oh[1], Seungjun Lee[1], Yeonghye Cho[1], Jhoon Kim[3], Wonjin Lee[4], Jaewoo Park[1,2], Ick Hoon Jin[1,2], and Sangwook Kang[1,2]

[1]Department of Statistics and Data Science, Yonsei University, Seoul, Republic of Korea
[2]Department of Applied Statistics, Yonsei University, Seoul, Republic of Korea
[3]Department of Atmospheric Sciences, Yonsei University, Seoul, Republic of Korea
[4]Environmental Satellite Center, National Institute of Environmental Research, Ministry of Environment

**Correspondence:** Jaewoo Park (jwpark88@yonsei.ac.kr), Ick Hoon Jin (ijin@yonsei.ac.kr), and Sangwook Kang (kanggi1@yonsei.ac.kr)

**Abstract.** This study presents advancements in the processing of satellite remote sensing data, focusing mainly on Aerosol Optical Depth (AOD) retrievals from the Geostationary Environment Monitoring Spectrometer (GEMS). The transformation of Level 2 (L2) data, which includes atmospheric state retrievals, into higher-quality Level 3 (L3) data is crucial in remote sensing. Our contributions lie in two novel improvements to the processing algorithm. First, we improve the inverse distance weighting

algorithm by incorporating quality flag information into the weight calculation. By assigning weights inversely proportional to the number of unreliable grids, the method can provide more accurate L3 products. We validate this approach through simulation studies and apply it to GEMS AOD data across various regions and wavelengths. The use of the quality flags in the algorithm can provide a more accurate analysis in remote sensing. Second, we employ a spatio-temporal merging method to address both spatial and temporal variability in AOD data, a departure from previous approaches that solely focused on spatial

variability. Our method considers temporal variations spanning previous time intervals. Furthermore, the computed mean fields show similar spatio-temporal patterns to the previous studies, confirming that they can capture real-world phenomena. Lastly, utilizing this procedure, we compute the mean field estimates for GEMS AOD data, which can provide a deeper understanding of the impact of aerosols on climate change and public health.

## 1 Introduction

In satellite remote sensing missions, observed data is processed at different levels. Using retrievals of the atmospheric state (Level 2; L2), L2 AOD products are regridded into Level 3 (L3) going through the process of filling gaps and filtering out noises (Cressie, 2018). We first introduce the theoretical background of the mean fields algorithm for generating L3 data application to aerosol optical depth (AOD) retrievals from the Geostationary Environment Monitoring Spectrometer (GEMS) satellite. We consider an oversampling method for generating L3 AOD data, inverse distance weighting (IDW), and a modified mean field

algorithm with consideration to spatio-temporal variability of AOD data in the algorithm.





Aerosols play a critical role in radiative forcing, climate change, and air quality (Brauer et al., 2015; Charlson et al., 1992; Stocker, 2014; Kaufman et al., 2002). Directly, they change the planetary albedo by reflecting solar radiation and absorbing terrestrial radiation, affecting the radiation balance. Indirectly, as cloud condensation nuclei, aerosols modify cloud properties and increase cloud droplet concentration, impacting solar radiation and cloud albedo (Alexander et al., 2013). Aerosol affects human health and air quality, especially in the regions affected by long-range transport or the regions with heavy aerosol emissions due to rapid industrialization and high population density. Those are linked to cardiovascular, respiratory, and allergic diseases, and even increased mortality rates (Pöschl, 2006; Tager, 2013).

Additionally, high aerosol concentrations can severely reduce visibility, leading to hazardous weather conditions like haze, smog, and dust storms (Charlson, 1969; Chen and Tsai, 2001). Thus, understanding aerosols' multifaceted impacts is crucial for addressing climate change, public health, and environmental visibility issues. The distribution of aerosols is characterized by their complexity, leading to increased uncertainty in determining aerosol radiative forcing effects (Chen et al., 2022). Analyzing aerosol's spatio-temporal distribution remains crucial for developing air pollution control policies and understanding the climate impacts of aerosols. Although accurate Aerosol Optical Properties (AOPs) and their vertical profiles can be obtained from ground-based measurements at the high temporal resolution, their AOPs can represent local-scale variability of limited spatial coverage. Unlike ground-based instruments, the regional and global monitoring of AOPs has been conducted by using satellite measurements.

A previous study of Park et al. (2023) focused on AOD retrievals by considering spatial variability. Specifically, Park et al. (2023) used the IDW algorithm to regrid L2 products and estimated the mean field of L3 products by considering spatial variability. Compared to the previous work, we have the following contributions. First, we have integrated quality flag information in the IDW algorithm so that we can rule out unreliable grid points. By considering variability in L2 AOD products, we can obtain more reliable L3 AOD products in this step. Second, we use the spatio-temporal merging method (Kikuchi et al., 2018) to obtain L3 AOD mean field estimates. We first compute spatio-temporal variabilities and use them to filter out uncertain values. We observe that our method can provide more realistic mean field estimates compared to the work Park et al. (2023) only considering spatial variability.

The outline of the remainder of this manuscript is as follows. In Section 2, we describe the GEMS data used in our analysis. In Section 3, we describe our method to compute the mean field of L3 AOD products. In Section 4, we conduct simulation studies to validate our method. We apply the proposed method to GEMS data in Section 5. We conclude with a discussion in Section 6.

## 2 GEMS Data

GEMS is the first UV-Vis hyperspectral satellite instrument onboard the Geostationary Korea Multi-Purpose Satellite-2B (GK-2B), launched on February 19, 2020. Its mission is to monitor air quality across Asia (5°S–45°N, 75–145°E) with high temporal (1-hour) and spatial resolution (3.5 × 7.7 km² at Seoul, South Korea), using hyperspectral measurements in the 300–500 nm range.



The GEMS aerosol retrieval (AERAOD) algorithm retrieves AOD, single scattering albedo (SSA), and aerosol layer height (ALH) using GEMS L1 data from six wavelengths (354, 388, 412, 443, 477, and 490 nm). This algorithm solves the limited degree of freedom for signal problems in the GEMS wavelength range by using two-channel inversion to retrieve the initial guesses of AOD and SSA and then inputting them into the optimal estimation method. This retrieval method was tested with sensitivity in the UV-Vis region to aerosol absorption and with Ozone Monitoring Instrument (OMI) Level 1 data (Kim et al., 2018; Go et al., 2020a, b). Initially developed from synthetic OMI data by Kim et al. (2018) and Go et al. (2020b), the operational version was later improved by Cho et al. (2023) based on real GEMS Level 1 data. An update to the aerosol algorithm, Version 2.0, was released in November 2022, which included reprocessing earlier data.

In this study, the following variables are used for calculating L3 AOD mean fields (Table 1).

**Table 1.** Description of the variables for GEMS L2 AERAOD data.

| Group | Variable |
|---|---|
| Data Fields | AOD at wavelength of 354nm, 443nm, 550nm |
| | 16-bit Quality Flag |
| Geolocation Fields | Longitude |
| | Latitude |
| | Solar Zenith Angle |
| | Viewing Zenith Angle |

Additional to Solar Zenith Angle (SZA) and Viewing Zenith Angle (VZA), due to the unavailability of cloud fraction in the GEMS AOD product, we utilize the GEMS L2 cloud product as masking criteria. The GEMS L2 cloud product is obtained from the GEMS cloud retrieval algorithm (Kim et al., 2024). With the same hyperspectral measurement range, temporal and spatial resolution as the AERAOD algorithm, the GEMS cloud retrieval algorithm retrieves the effective cloud fraction (ECF) and provides the cloud radiation fraction (CRF) via the CRF conversion process (Choi et al., 2020). To filter out pixels biased with high cloud fraction, we leverage CRF with wavelength corresponding to AOD product

## 3   Methodology

We apply a three-stage procedure to calculate the mean field of the L3 AOD products. First, we regrid L2 AOD products using the IDW method with neighboring spatial information to obtain the L3 AOD products. Then merge the L3 AOD products by considering spatio-temporal variability in the products according to Kikuchi et al. (2018). Specifically, we merge the L3 AOD products using the previous $T$ time products from the target products of interest. Lastly, we produce the mean field of the L3 AOD products by taking a simple mean of the spatio-temporally merged L3 data. The outline of the method is illustrated in Figure 1.



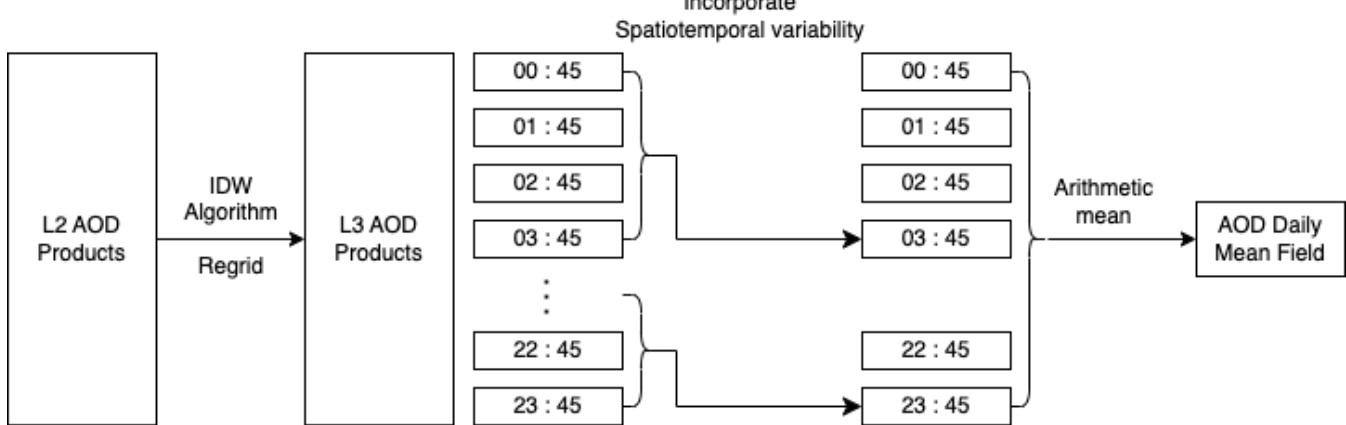

**Figure 1.** Illustration of the proposed estimation of the mean field methods under the window size of $T = 3$.

### 3.1 Inverse Distance Weighting

In this section, we describe the inverse distance weighting (IDW) algorithm that can obtain L3 AOD products. Several methods, including the nearest neighbor method (Lotrecchiano et al., 2021), the linear interpolation method (Abdullah et al., 2019; Shepard, 1968) and the spline interpolation method (Kuhlmann et al., 2013), have been proposed to interpolate the air quality

mass. The IDW algorithm (Zimmerman et al., 1999) is one of the most popular methods among the linear interpolation methods due to its computational simplicity. Our goal is to obtain the L3 AOD products for each longitude-latitude location.

Let $(x_0, y_0)$ be the target longitude-latitude location for calculating the L3 AOD product. Suppose $(x_1, y_1), \cdots, (x_n, y_n)$ represent the neighboring longitude-latitude locations to $(x_0, y_0)$, each paired with its respective L2 AOD product denoted by $\mathrm{AOD}(x_1, y_1), \cdots, \mathrm{AOD}(x_n, y_n)$. To calculate the L3 AOD product in our application, we use grid points within the $r$-radius of

$(x_0, y_0)$. This means that, for the given $(x_0, y_0)$, we use the locations that satisfy $x_i \in (x_0 - r, x_0 + r)$ and $y_i \in (y_0 - r, y_0 + r)$. Specifically, we set the radius $r$ at 0.1 ° for the East Asia region and 0.05° for the Korean Peninsula region. Then, for the fixed observed time point $t_0$, the IDW estimate is

$$\mathrm{AOD}_{IDW}(x_0, y_0, t_0) = \sum_{i}^{n} \lambda_i \mathrm{AOD}(x_i, y_i, t_0), \tag{1}$$

where the weight of each location is defined as

$$\lambda_i = \frac{1/d_i^p}{\sum_i^n 1/d_i^p}. \tag{2}$$

In Equation (2), $d_i$ is the Euclidean distance from $(x_0, y_0)$ to $(x_i, y_i)$ and $p$ is the power parameter. Therefore, Equation (1) is based on the weighted average of the L2 AOD values from neighboring locations; the larger weight is assigned to grid points close to the location of interest $x_0$.

Depending on the choice of $p$, the IDW estimates yields different outcomes. As $p$ goes to 0, Equation (2) becomes equal

weight and the IDW estimate gets close to a simple average from neighboring locations. On the other hand, as $p$ goes to $\infty$,



the larger weight is concentrated on the locations near $x_0$ and the IDW estimate converged to the estimate obtained through the nearest neighbor. Although the optimal choice of $p$ can be different depending on the study (Liu et al., 2006), $p = 2$ is the most commonly used value since when $p = 2$, the weight for distance between grid points decays relatively faster to avoid the discontinuity (Webster and Oliver, 2007). Following this convention (Isaaks and Srivastava, 1989), we also use $p = 2$ in our analysis.

### 3.1.1 An Enhanced Inverse Distance Weighting with Quality Flag Information

A quality flag is an indicator that contains data quality information for each grid points. Such an indicator is widely used for data cleaning and selection. In our study, we have quality flag information in the L2 AOD products (coded as a 16-bit unsigned integer value). The quality flag used in our study is described in Table 2.

**Table 2.** Quality flags information.

| Bits | Definition | Note | Description |
|---|---|---|---|
| 0 | Reliable | Good | (0, Good; 1: have issue)  AOD >0.2 & ALH AK >0.2 |
| 1 | Less Reliable | Suspect | AOD <0.2 or ALH AK >0.2 |
| 2 | Out of bounds SSA or AOD at 443 nm. | Bad | AOD <-0.05 or AOD >3.6 or SSA <0.82 or SSA >1.0 |
| 3 | OE fitting error | Bad | Fitting error during optimal estimation |
| 4 | Normalized radiance above threshold | Bad | High normalized radiance |
| 5 | Surface albedo above threshold | Bad | High surface albedo |
| 6 | Cloud masking | Cloud | Presence of clouds |
| 7 | Solar zenith angle above threshold (69°) or viewing zenith angle above threshold | Bad | SZA >69° or VZA >69° |
| 8 | Sun-glint angle below threshold over water | Bad | Sun glint angle <35° |
| 9 | Terrain height high | Suspect | Terrain height >35° |
| 10 | Previous L2 SFC (-5 day) are used | Suspect | Absence of L2 SFC information |
| 11 | OMI climatology used for surface albedo | Suspect | Absence of L2 SFC information |
| 12 | Previous irradiance used | Suspect | Absence of L1C irradiance |
| 13 | AMI cloud-masking used | Cloud | Cloud masking using AMI L2 Cloud product |
| 14 | Less reliable of surface albedo | Suspect | Less accurate AERAOD surface albedo |
| 15 | Interpolated radiance used | Suspect | LIC Radiance QF=2 |

To incorporate quality flag information in the analysis, we convert a 16-bit integer to a binary value. For instance, the number 196 can be expressed as 000000011000100, which implies the features of bit 2, bit 6, and bit 7 are contained. According to Table 2, pixels with an algorithmic quality flag of 196 are likely to have features of a smaller AOD value than -0.05 or a larger value than 3.6. In addition, it may have a smaller SSA value than 0.82 or larger than 1.0 in the presence of clouds and solar zenith angle being above the threshold or viewing zenith angle being above the threshold. Table 3 provides further details about the quality flags.





**Table 3.** Process of converting algorithm quality flag 196 into 16-bit unsigned integers to binary. The first equation shows that the decimal number 196 can be summed as $196 = 2^7 + 2^6 + 2^2$, which indicates that it can be converted to the binary number 000000011000100 as written in the second line. To show which bit has the issue based on the converted binary number, we enumerate the bit 0 to bit 16 on the third line. The process shows that bit 2, bit 6, and bit 7 have an issue.

| 196 = | | | | | | | | $2^7$ | $2^6$ | | | | $2^2$ | | |
|---|---|---|---|---|---|---|---|---|---|---|---|---|---|---|---|
| 196 = | 0 | 0 | 0 | 0 | 0 | 0 | 0 | 1 | 1 | 0 | 0 | 0 | 1 | 0 | 0 |
| bit = | 15 | 14 | 13 | 12 | 11 | 10 | 9 | 8 | 7 | 6 | 5 | 4 | 3 | 2 | 1 | 0 |

Then, we define an uncertainty metric $u_i$ corresponding to a weight $\lambda_i$ used in the IDW method. The calculation of uncertainty metric, denoted as $u_i$, is based on a quality flag that is represented by a 16-bit unsigned integer. As mentioned, this integer is first converted into binary format. We then add all the problematic bits with a value of 1 to compute $u_i$. With this quality flag information, the IDW weight used for our method is

$$\lambda_i = \frac{1/d_i^p u_i^q}{\sum_i 1/d_i^p u_i^q}, \tag{3}$$

where $u_i = \sum (\text{bit values of the qualify flag}) + 1$. Since the high values of $u_i$ imply the low quality of the data, we take the inverse of $u_i$ in the weights of the IDW algorithm. In Equation (3), $q$ is a power parameter that controls the amount of quality flag information. The larger the $q$ value, the higher penalty will be assigned to the grid with a large $u_i$ value. In Section 4, we observe that incorporating quality flags in the weight can improve the accuracy of the IDW method. Furthermore, we validate the choice of quality flags from the simulation study.

## 3.2 Spatio-temporal Merging Algorithm

In this section, we describe a merging algorithm (Kikuchi et al., 2018) that can account for spatio-temporal variability in L3 AOD products. By using spatio-temporal information, we can adjust the weights to produce a more robust and accurate L3 AOD mean field output.

### 3.2.1 Spatio-temporal Variability of AOD$_{IDW}$

It is crucial to consider the spatio-temporal variability of the IDW estimates when we compute the mean field of L3 AOD products (Kikuchi et al., 2018). However, we only have a single IDW estimate, AOD$_{IDW}(x, y, t)$ at a specific location and time. Since we do not have repeated measures of AOD$_{IDW}(x, y, t)$, the spatio-temporal variability should be computed using neighboring information. Let $(x_0, y_0, t_0)$ be the location and time of interest and $(x_i, y_i, t_i)$ be its neighboring location $i$. Then spatio-temporal variability is defined as a root-mean-square difference (RMSD) of AOD$_{IDW}$ estimates as

$$\sigma_{IDW}(x_0, y_0, t_0) = \sqrt{\frac{1}{N} \sum_i^N (AOD_{IDW}(x_i, y_i, t_i) - AOD_{IDW}(x_0, y_0, t_0))^2} \tag{4}$$



where $N$ is the number of neighboring pixels within the radius of $r$ and previous $T$ time from $(x_0, y_0, t_0)$. In this work, we consider the radius of four grid lengths and window size of $T = 3$ so that $r = 0.1°, 0.2°, 0.3°, 0.4°$ and $t = 0, 1, 2, 3$.

### 3.2.2 Hourly Merged AOD Estimates

Using the spatio-temporal variability in Section 3.2.1, we computed hourly combined AOD products. The procedure is summarized as follows. First, we obtain $AOD_{pure}$ by filtering out unreliable grid points. Then, we compute $AOD_{merged}$ at the location of interest by interpolating $AOD_{pure}$. From this, we can retrieve reliable AOD products.

**Computing $AOD_{est}$**

We first introduce $AOD_{est}(x_0, y_0, t_0)$, which is a weighted average of the IDW estimates obtained in Section 3.1. The AOD
estimate at a target grid $(x_0, y_0, t_0)$ is

$$AOD_{est}(x_0, y_0, t_0) = \sum_{i}^{n} w_i AOD_{IDW}(x_i, y_i, t_0), \tag{5}$$

where

$$w_i = \frac{\frac{1}{\sigma^2_{IDW}(x_i, y_i, t_0)}}{\frac{1}{\sigma^2_{est}(x_i, y_i, t_0)}} \text{ and } \frac{1}{\sigma^2_{est}(x_0, y_0, t_0)} = \sum_{i}^{n} \frac{1}{\sigma^2_{IDW}(x_i, y_i, t_0)}. \tag{6}$$

Here, $n$ denotes the number of effective pixels within the spatial radius $r$ and past $T$ time from $(x_0, y_0, t_0)$, whose $AOD_{IDW}$
values are greater than equal to 0. In Equation (5), $AOD_{est}$ is the weighted average of $AOD_{IDW}$ and weights are defined by the inverse of the spatio-temporal variability in Section 3.2.2. Note that the inverse of the variability quantifies the accuracy and reliability of the IDW estimate at each grid; $w_i$ implies the sum of accuracies over the neighboring region of the target point.

**Estimating the Error Variance**

Our goal is to filter out grid points with high variability. Note that the spatio-temporal variability in Equation (4) becomes small
as the spatial or temporal distance between grids becomes larger. Utilizing this relationship, we estimate spatial and temporal variability separately through a regression model. The combined variability, denoted as $\sigma_0$, at the currently considered grid point $(x_0, y_0, t_0)$ is then calculated as the mean of the spatial and temporal variabilities.

  Before estimating the variability, we categorize the value of $AOD_{IDW}$ with different classes. This is because the pattern of spatio-temporal variability varies depending on the magnitude of AOD values (Kikuchi et al., 2018). Specifically, we categorize
$AOD_{IDW}$ values into 6 bins of 0.1, 0.25, 0.5, 0.75, 0.9, and 1.0. Note that the previous work (Kikuchi et al., 2018) used 12 number of classes. On the other hand, we use 6 classes because certain classes are rarely observed; using 12 classes can lead to unreliable computation.

  Let $\sigma_{dist}(x_0, y_0, t_0)$ be the spatial variability and $\sigma_{time}(x_0, y_0, t_0)$ be the temporal variability of the IDW estimate at $(x_0, y_0, t_0)$. We first compute the average of spatio-temporal variability $\sigma_{IDW}(x_i, y_i, t_i)$ by radius and class. Then we regress
the averaged values obtained for each component of the radius vector, $r = (0.1°, 0.2°, 0.3°, 0.4°)$, on a design matrix $[1, r, r^2]$,





which is a second-order design matrix of radius $r$ for each class. Lastly, we obtain a spatial variability $\sigma_{dist}(x_0, y_0, t_0)$ from the intercept estimate of the quadratic regression model fitting. We can obtain a temporal variability $\sigma_{time}(x_0, y_0, t_0)$ in the similar manner. We first compute the average of spatio-temporal variability $\sigma_{IDW}(x_i, y_i, t_i)$ by time point and class. We regress them on a design matrix $[1, t, t^2]$ for each class and obtain $\sigma_{time}(x_0, y_0, t_0)$ from the intercept estimate of the quadratic regression model fitting.

Finally, we compute the error variance by taking the average of $\sigma_{dist}$ and $\sigma_{time}$ as

$$\sigma_0(x_0, y_0, t_0) = \frac{\sigma_{dist}(x_0, y_0, t_0) + \sigma_{time}(x_0, y_0, t_0)}{2}. \tag{7}$$

The calculated error variance contains measurement error caused by sensor noise that varies over time and space shift.

**Computing $AOD_{pure}$**

Here, we obtain $AOD_{pure}$ by filtering out uncertain $AOD_{IDW}$ values. For this, we introduce the estimated error of $AOD_{pure}$, which is defined as

$$\sigma_{pure}(x_0, y_0, t_0) = \begin{cases} \sqrt{\sigma_0^2(x_0, y_0, t_0) + \sigma_{est}^2(x_0, y_0, t_0)}, & \text{if } AOD_{IDW}(x_0, y_0, t_0) \text{ is observed,} \\ \text{missing,} & \text{otherwise.} \end{cases}$$

Here, $\sigma_{est}^2$ is the error variance of $AOD_{est}$ from Equation (6) and $\sigma_0^2$ is the combined error variance from Equation (7). To filter out uncertain $AOD_{IDW}$ values, we consider an upper threshold of $AOD_{IDW}$ as

$$AOD_{pure}(x_0, y_0, t_0) = \begin{cases} AOD_{IDW}(x_0, y_0, t_0), & \text{if } AOD_{IDW}(x_0, y_0, t_0) \leq AOD_{est}(x_0, y_0, t_0) + 2.58\sigma_{pure}(x_0, y_0, t_0), \\ \text{missing,} & \text{otherwise.} \end{cases} \tag{8}$$

Assuming Gaussian distribution, if $AOD_{IDW}(x_0, y_0, t_0)$ exceeds the upper threshold of the 99% confidence interval, we consider the value is not reliable and exclude it from the mean field calculation.

**Computing $AOD_{merged}$**

Based on section 3.2.1, we calculate the total variability within the neighbor grids of the target grid. Then, we use the ratio of the inverse of this total variability as weights to calculate a weighted average for $AOD_{pure}$, resulting in $AOD_{merged}$ as follows:

$$AOD_{merged}(x_0, y_0, t_0) = \begin{cases} \sum_i^n w_i AOD_{pure}(x_i, y_i, t_0), & \text{if } AOD_{IDW}(x_0, y_0, t_0) \text{ is observed,} \\ \text{missing,} & \text{otherwise,} \end{cases} \tag{9}$$

where

$$w_i = \frac{\frac{1}{\sigma_{pure}^2(x_i, y_i, t_0)}}{\frac{1}{\sigma_{merged}^2(x_0, y_0, t_0)}} \text{ and } \frac{1}{\sigma_{merged}^2(x_0, y_0, t_0)} = \sum_i^n \frac{1}{\sigma_{pure}^2(x_i, y_i, t_0)}.$$





This merging process not only utilizes the reliable value $\text{AOD}_{pure}$ but also incorporates the reliability $\sigma_{pure}$ as a weight, resulting in a more trustworthy gap-filling outcome. In fact, in the study by Kikuchi et al. (2018), the RMSE of $\text{AOD}_{merged}$ was notably lower at 0.11 compared to the RMSE of 0.20 for $\text{AOD}_{IDW}$.

## 4 Simulation

### 4.1 Generating Simulation Data

In this section, we conduct a simulation study to validate the choice of quality flags. The data generation procedure is summarized as follows.

1. We generate an each element of $\mathbf{X} \in \mathbb{R}^{4900 \times 2}$ from $U(0,1)$ and use $\boldsymbol{\beta} = (1,1)$ as a true coefficient value.

2. We create a $70 \times 70$ lattice over the $[0,1]^2$ domain and regard each grid as an observed location of the dataset. Then we simulate zero-mean Gaussian process $\boldsymbol{W}$ from $N(0,(\tau \mathbf{M}^{'}\mathbf{QM})^{-1})$ where $\mathbf{M}$ is obtained by taking the first k eigenvectors

of the Moran operator (Hughes and Haran, 2013) with smoothness parameter $\tau$. Here, $\mathbf{Q} = \text{diag}(\mathbf{A1})$ - $\mathbf{A}$ is a precision matrix calculated from the adjacency matrix, $\mathbf{A}$.

3. We simulate AOD datasets from $\boldsymbol{Y} = \boldsymbol{X}\boldsymbol{\beta} + \boldsymbol{W} \in \mathbb{R}^{4900 \times 2}$. In our simulation, $\boldsymbol{X}\boldsymbol{\beta}$ represents the fixed effect, while $\boldsymbol{W} \in \mathbb{R}^{4900}$ can account for spatial correlation in AOD products.

4. We generate missing data for the simulated $\boldsymbol{Y}$ as in GEMS L2 product. For realistic simulation, we apply the observed

missing pattern in the GEMS AOD data to the simulated $\boldsymbol{Y}$ in Step 3. Specifically, we use the missing pattern of 440nm observed on April 1st 04:45 in the partial area of East Asia, which contains about 20% of missing values.

We repeat Steps 1 - 3 for 100 times to generate different realizations of $\boldsymbol{Y}$. Then, for each simulated $\boldsymbol{Y}$, we apply the identical missing patterns (Step 4) obtained from the real dataset. Figure 2 illustrates an example of a simulated dataset.

### 4.2 Sensitivity Analysis of $q$

Here, we investigate the performance of the IDW method on GEMS data by varying $q$ in Equation (3). Specifically, we consider $q = 0.5, 1$, and $1.5$ in our experiment. We first examine whether there is a significant difference in IDW estimates with difference choices of $q$. Figure 3 indicates that the IDW estimates are comparable with different $q$ values. Table 4 also shows that the summary statistics of $\text{AOD}_{IDW}$ are quantitatively similar with different $q$ choices. Therefore, we conclude that the IDW algorithm is robust to the choice of $q$. To simplify the calculation, we set $q = 1$ in our analysis.

### 4.3 Quality Flag Simulation

As explained in section 4.2 and Equation (3), quality flag indicators weigh the uncertainty of the IDW algorithm. To improve the accuracy of the IDW algorithm, it is necessary to find the optimal bit combination of the quality flag by performing simulation





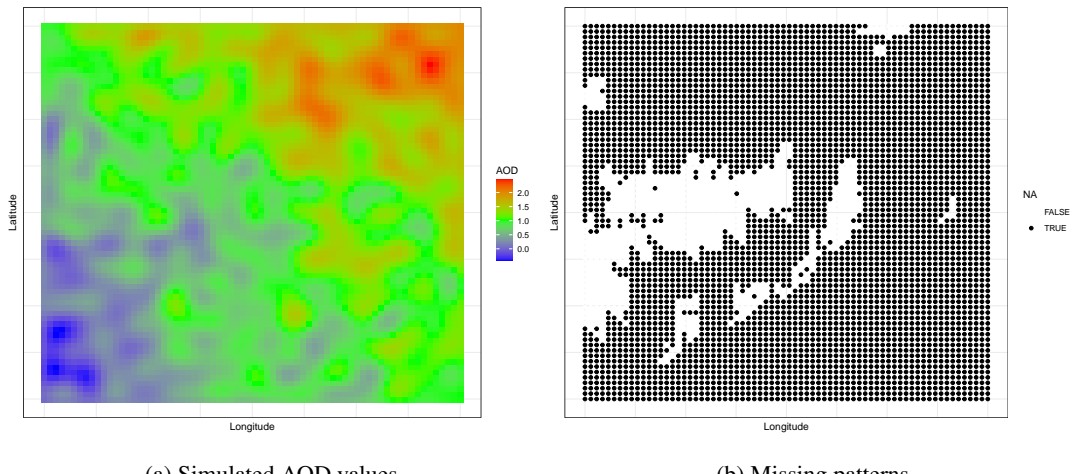

(a) Simulated AOD values                (b) Missing patterns

**Figure 2.** The left figure illustrates the simulated AOD dataset and the right figure shows the missing pattern (white color).

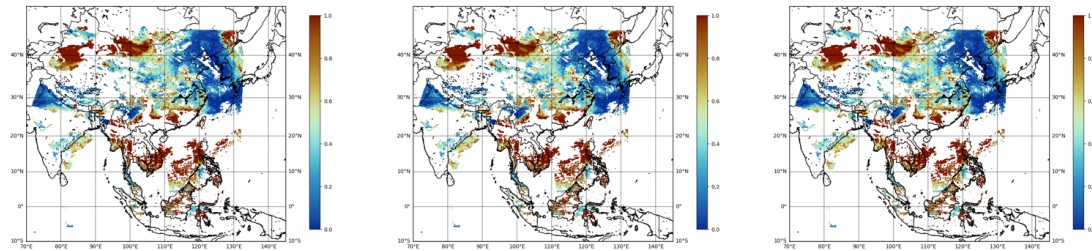

**Figure 3.** Comparison of the IDW estimates on April 1, 2023, at 7:00 am. Each figure illustrates $\text{AOD}_{IDW}$ with $q = 0.5$ (left), $q = 1$ (middle), and $q = 2$ (right).

studies. Therefore, we first discover bits that show substantially lower MSE, then combine such particular bits into groups. Note that we refer to these bit combinations of the quality flags as a "case." For example, we may discover bit 0,1,2 has a

215 significantly low MSE. Then we can create various combinations that include 0,1,2 such as $0 \cdot 1$, $0 \cdot 2$, or even $0 \cdot 1 \cdot 2$, and each can be denoted as a particular "case".

**Table 4.** Summary statistics of $\text{AOD}_{IDW}$ values with different $q$ values on April 1, 2023, at 7:00 am.

| Exponent ($q$) | Mean | Standard deviation | Min | 25% quantile | Median | 75% quantile | Max |
|---|---|---|---|---|---|---|---|
| $q = 0.5$ | 0.6139 | 0.5534 | 0.0 | 0.2003 | 0.4895 | 0.8654 | 3.5928 |
| $q = 1$ | 0.6121 | 0.5539 | 0.0 | 0.1966 | 0.4876 | 0.8642 | 3.5928 |
| $q = 2$ | 0.6173 | 0.5524 | 0.0 | 0.2081 | 0.4935 | 0.8673 | 3.5928 |



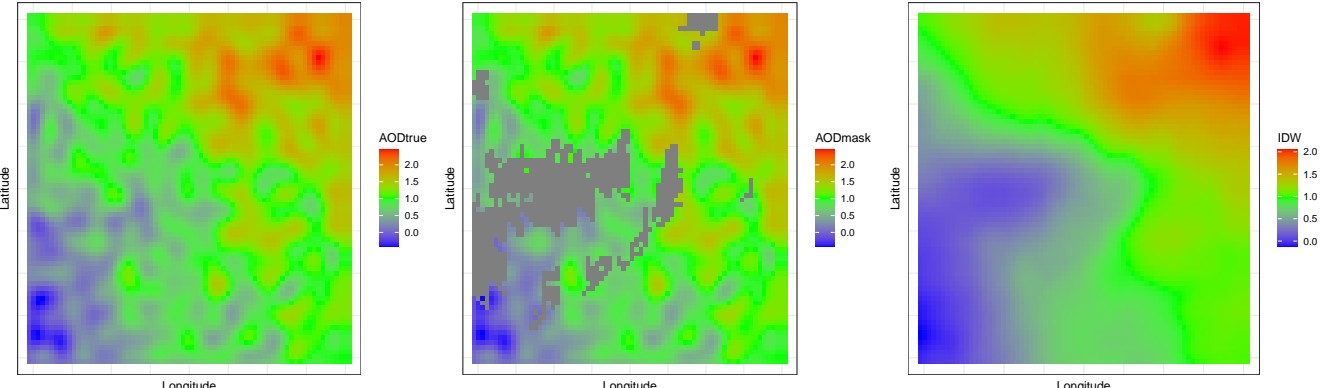

**Figure 4.** The left figure shows the simulated AOD values while the middle figure shows the simulated data incorporated with the missing pattern, which can be distinguished by the gray color. The right figure illustrates the result of applying the IDW algorithm to simulated data with radius 9.

Before finding an optimal case of the quality flag through a simulation study, we examine whether the result differs depending on the radius of the IDW algorithm $r$. Note that this simulation study defines the unit of $r$ radius as one unit grid, not the distance based on coordinates as described in 3.1. We fixed the radius as 9 since varying the radius size did not show any notable difference in MSE, yet it shows a better imputation of the IDW algorithm as described in Appendix A. Figure 4 shows the generated AOD simulation data, where the center figure is the data applying the missing value pattern, and the right figure is the visualization result showing the result of applying the IDW algorithm to the data.

We then follow the following procedure. After selecting particular bits with lower MSE than others fixed with radius 9, we repeat the experiment to reflect diverse uncertainty term calculations by considering various cases in the IDW algorithm. We then find the optimal case that obtains the highest accuracy after comparing the accuracies between the cases. MSE values are evaluated between the simulation and the imputed data after regarding the simulated data as real data.

To select the quality flag bit for making the combination case, we first calculate the MSE for every quality flag bit. Figure 5 expresses the MSE value as a boxplot for every bit. Although each consecutive boxplot indicates the result of bit 0 to 15, bit 2 and bit 6 have significantly lower MSEs than others. The medians for each bit are 0.122 and 0.123, respectively. Bit 2 defines whether SSA or AOD is out of a specific value range, while bit 6 shows the presence of clouds. We then compose six various candidate cases, including bit 2 and bit 6. Table 5 summarized six various candidate cases.

To find the optimal quality control case that yields the lowest MSE among six candidate cases, we calculate the MSE for each case with two different $\tau$, $\tau = 1$ and $\tau = 6$, and draw boxplots in 6. From Figure 6(a) and Figure 6(b), each case shows a different MSE, while Case 5, which contains bit 0, 2, and 6, shows the lowest median of 0.387 in Figure 6(a). Therefore, in our real application, we use the quality flag to Case 5 (bit 0, 2, and 6) when calculating the uncertainty term for the IDW algorithm.

We check whether the optimal case for IDW algorithm changes depending on the smoothing parameter $\tau$ of simulation data. Figure 6(a) and Figure 6(b) show the average MSE, which is represented as a box plot as above, with different smoothing





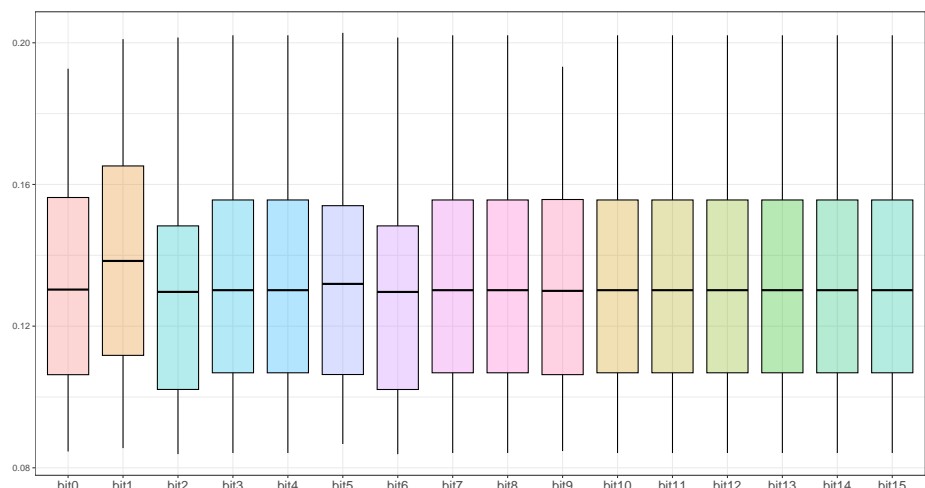

**Figure 5.** Boxplot describing the IDW algorithm results on 100 repetitive simulation datasets, including bit 0 to bit 15 respectively in uncertainty metric $\sigma(i)$. We discover that bit 2 and bit 6 have lower MSE compared to other bits.

**Table 5.** Six candidate cases for selecting quality flag with the combinations of bit 0, 2, and 6.

|      | Case 1 | Case 2 | Case 3 | Case 4 | Case 5 | Case 6 |
|------|--------|--------|--------|--------|--------|--------|
| Bits | All    | 0      | 2      | 6      | 0,2,6  | Nothing |

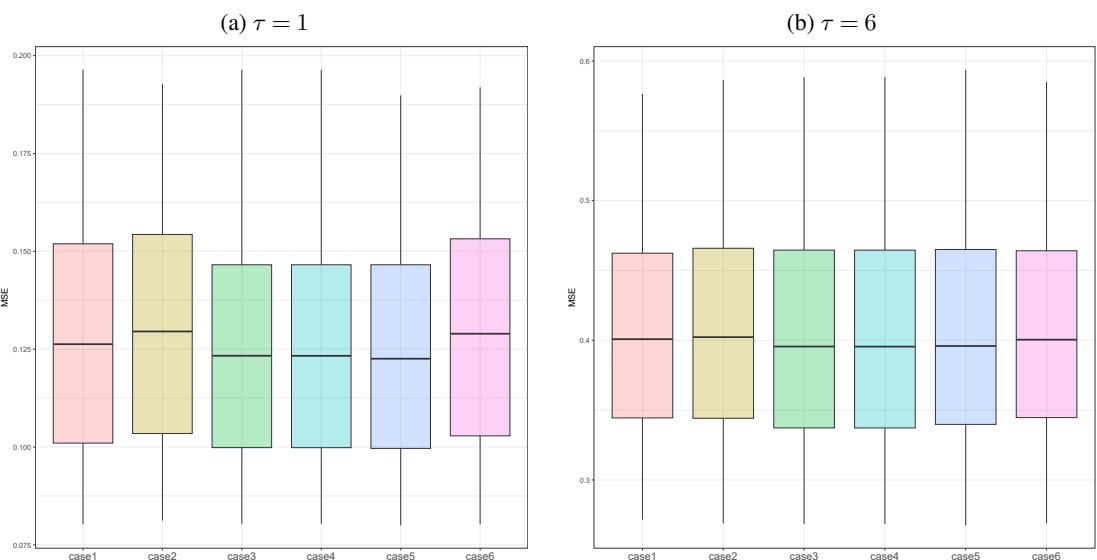

**Figure 6.** The left figure shows the simulation results with $\tau = 1$ while the right figure shows the results with $\tau = 6$. Despite the value of smoothing parameter $\tau$, we discover that the hierarchy of the MSE values between the cases are identical.





parameters. We discover that the overall magnitude of the MSE value remains different, but the hierarchy of the MSEs across the cases is unchanged. It implies that the algorithm's performance is robust regardless of the smoothing parameter.

## 5  GEMS Data Application

### 5.1  Spatial Resolution of Output Product

In our study, we set the mean field spatial resolution as $0.1° \times 0.1°$ longitude-latitude grid, which is larger than the grid length of 7 km × 8 km. A geostationary satellite such as GEMS observes a fixed position. Therefore, if we use grids that are too small, the output will have many missing values; an appropriate grid size should be selected in terms of the coverage of the mean fields. Furthermore, we need to consider effective range, accuracy, and computation time to obtain mean fields when we choose a grid size. If we use too fine spatial resolution, computational cost will be exponentially increased. On the other hand, grid sizes that are too large can lead to inaccurate output results.

### 5.2  Level 2 Aerosol Optical Depth Data

As evidenced by various studies (Kaufman et al., 2005; Loeb and Manalo-Smith, 2005; Matheson et al., 2005), AOD exhibits a positive correlation with Cloud Fraction (CF), implying that proximity to clouds can result in a statistical increase in AOD measurements. Additionally, a dependency of AOD on solar and viewing zenith angles was observed (de Miguel et al., 2011), highlighting the complexities involved in accurate AOD estimation under varied atmospheric conditions. To address these complexities, this study involved masking data based on quality flags incorporated within the GEMS AOD L2 product. These quality flags are meticulously designed to account for variability and uncertainties in satellite data (Choi et al., 2020), addressing factors such as solar and viewing zenith angles, and cloud fraction. This methodology ensures a more precise and reliable estimation of AOD, which is crucial in understanding atmospheric dynamics and environmental monitoring. The unreliable values are treated as missing values when we apply the IDW algorithm.

- cloud fraction $\geq 0.4$,

- solar zenith angle $\geq 70$ degrees, and

- viewing zenith angle $\geq 70$ degrees

In our study, we set two spatial domains of the mean field outputs, one with latitude (30° N-43° N) and longitude (123° E-131° E) corresponding to the vicinity of the Korean Peninsula and the other with latitude (32° N-43° N) and longitude (115° E-131° E) including the vicinity of the Shandong Peninsula in China.




## 5.3 Mean Field Estimates of GEMS AOD L3 Product

As we described in Section 3, we obtain $AOD_{pure}$ from $AOD_{est}$ by considering both the spatio-temporal variability $\sigma_{est}$ and the estimation uncertainty $\sigma_{est}$. This procedure allows us to obtain more reliable AOD estimates. Furthermore, we can use more robust AOD estimates when we compute the mean field, resulting in smoother output.

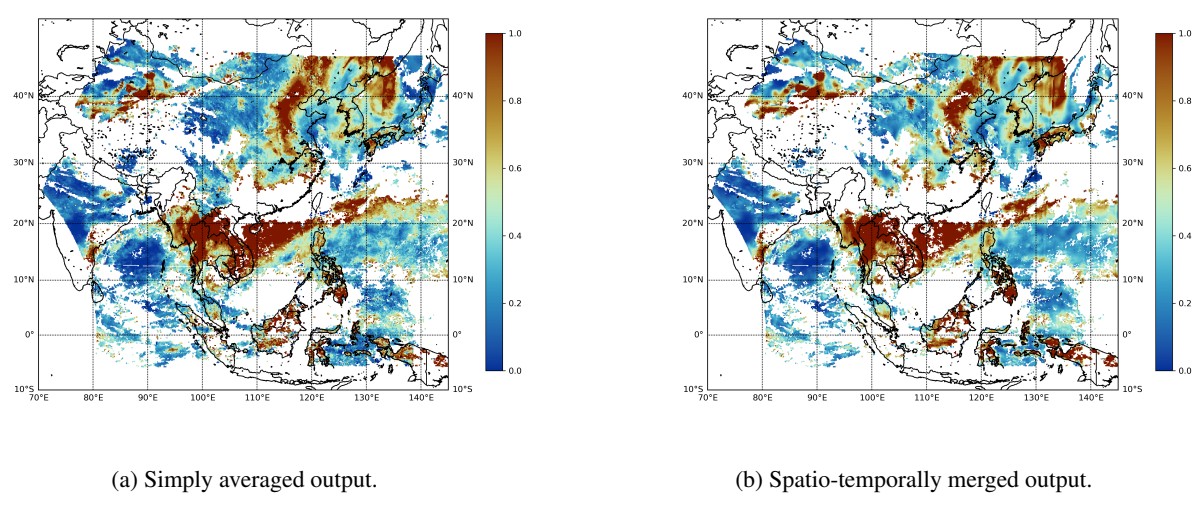

(a) Simply averaged output.  (b) Spatio-temporally merged output.

**Figure 7.** Daily mean-field estimates of GEMS AOD L3 product on April 1, 2023, at the wavelength of 354nm.

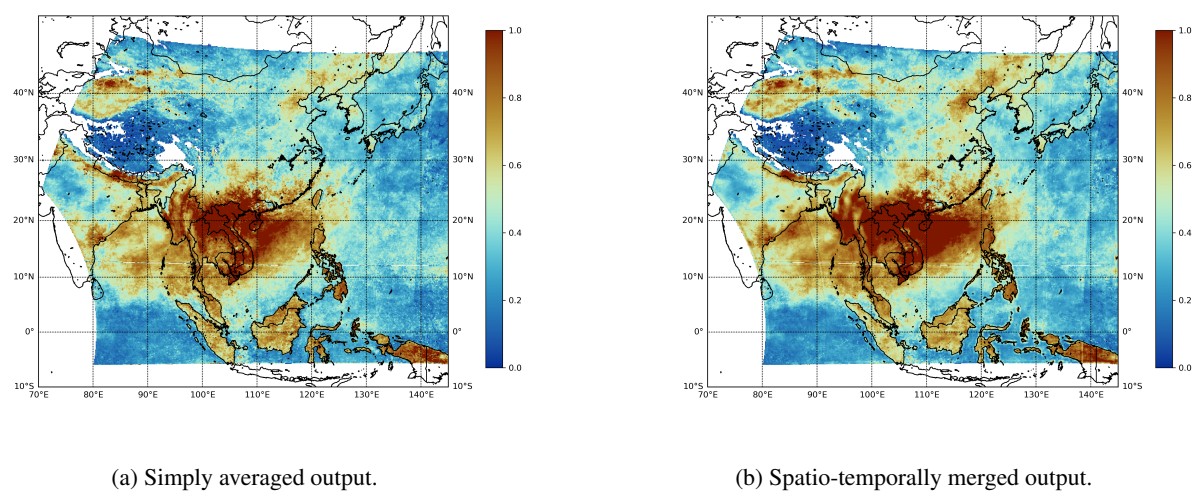

(a) Simply averaged output.  (b) Spatio-temporally merged output.

**Figure 8.** Monthly mean-field estimates of GEMS AOD L3 product in April, 2023, at the wavelength of 354nm.




Figures 7 and 8 compare the spatio-temporal merged products with the simply averaged products. For spatio-temporal merged AOD L3 products, we apply the procedure described in Section 3. On the other hand, we take a mean of the IDW estimates for simply averaged AOD L3 products. As mentioned in Kikuchi et al. (2018), grid points with AOD values exceeding 1.0 are extremely rare. Therefore, to focus on the majority of values for detailed characterization, we set the threshold at 1.0. In Figure 7, we observe that more missing values occur in the area of latitude (20° N-30° N) and longitude (130° E-140° E) for the spatio-temporally merged products compared to the simply averaged products. This is due to the fact that a spatio-temporal merging procedure only considers reliable AOD estimates, while a simple averaging method does not. Therefore, the simply averaged products can be regarded as more unreliable, though they have less number of missing values. In Figure 7, we also observe that the spatio-temporal merging method can provide smoother mean field outputs; for example, there is a significant difference in the area of latitude (35° N-45° N) and longitude (125° E-135° E). Similar trends are also observed in Figure 8. Mean field estimates at all three wavelengths for East Asia and Korea are provided in Appendix B and C, respectively.

## 5.4 Qualitative Evaluation for the Mean Field Products

Direct evaluation of the accuracy of the AOD L3 mean field products is challenging because there are no true values for the products. Therefore, we compare our results with previous studies with qualitative aspects.

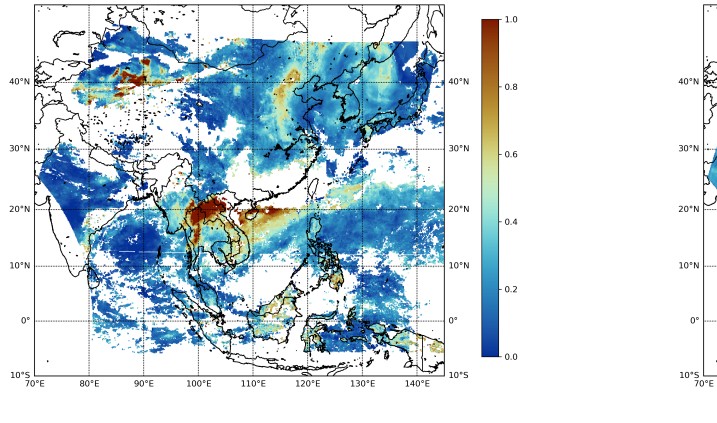
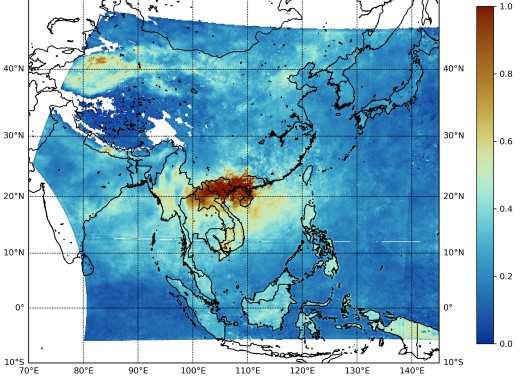

(a) Daily Mean Field Output on April 1, 2023.    (b) Monthly Mean Field Output on April 1, 2023.

**Figure 9.** Mean-field estimates of GEMS AOD L3 product on April 1, 2023 at the wavelength of 550nm.

In Figure 9 (a), we observe that our mean field products at 550nm wavelength on April 1 are similar to the springtime global distribution of AOD at the same wavelength. Furthermore, high values are observed near the Taklamakan Desert due to dust and also observed in Southeast Asia during spring due to biomass burning. This indicates that the computed mean fields can effectively capture real-world phenomena. In addition, we observe that overall trends of AOD values are similar to that from





MODIS data (Tian et al., 2023), though there are some discrepancies in the vicinity of the Taklamakan Desert and certain areas in Southeast Asia.

## 6    Conclusions

In remote sensing, data have been processed to different units. For example, the L2 dataset, which contains atmospheric state retrievals, is converted to the L3 dataset. Specifically, we focus on AOD retrievals from the GEMS satellite through gap filling and noise filtering. We improve the quality of L3 AOD mean field products by considering quality flag information and spatio-temporal variability (Kikuchi et al., 2018). Specifically, the contribution of our work is summarized as follows.

First, we improve the performance of the IDW algorithm by including quality flag information in the weight calculation. We assign weights inversely proportional to the number of poor quality indicators. To validate the choice of different quality flags, we conduct simulation studies. We observe that including bits 0, 2, and 6 from the quality flags significantly improves the accuracy of the IDW algorithm. We apply this novel approach to GEMS ADO data covering various regions and wavelengths.

Second, we apply spatio-temporal merging method (Kikuchi et al., 2018) to GEMS AOD data. Compared to the previous work (Park et al., 2023) that only considers spatial variability, our method can also account for temporal variability from the previous time points. We observe that our mean field products show a similar trend to the previous studies, indicating that the products are reliable.

Although our current study has made notable progress in enhancing the accuracy of AOD mean field estimation, several avenues for future research remain open. One potential direction involves integrating additional data, such as cloud information and the distinction between oceanic and terrestrial regions, which could further refine our results by considering the impact of cloud cover on aerosol retrievals. Additionally, validating our AOD mean field products against ground-based measurements or other satellite datasets could offer valuable insights into their reliability and consistency, thereby helping to identify any potential biases or uncertainties. Lastly, sensitivity analysis for the choice of hyperparameters (e.g., radius, time windows) would be useful to improve the performance of the method.

*Data availability.*    The GEMS Level 2 products are available at https://nesc.nier.go.kr/ko/html/index.do (last access: 28 February 2024)

## Appendix A:  Sensitivity Analysis for the Raidus of the IDW Algorithm

We compare the IDW algorithm application result varying the radius $r$. We find a remarkable difference in the imputed area viewed in the visualization as fewer missing values remain when the $r = 9$ compared to the $r = 3$. However, each MSE value is 0.255 and 0.25, respectively, showing no significant difference in numbers. Even with a negligible difference between the two window sizes, we determined that the window size should be 9 since it covers more missing areas and has a lower MSE value.





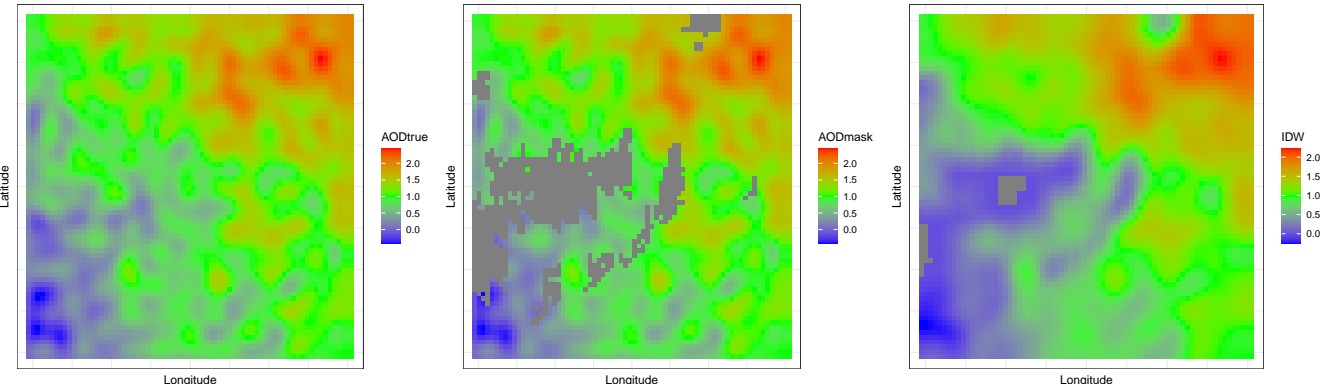

**Figure A1.** The left figure shows the simulated AOD values while the middle figure shows the simulated data incorporated with the missing pattern, which can be distinguished by the gray color. The right figure illustrates the result of applying the IDW algorithm to simulated data with radius 3.

## 315 Appendix B: Mean Field Estimates of GEMS AOD L3 Product for East Asia

In this section, we include daily and monthly mean-field estimates of GEMS AOD L3 products at all three wavelengths for East Asia.

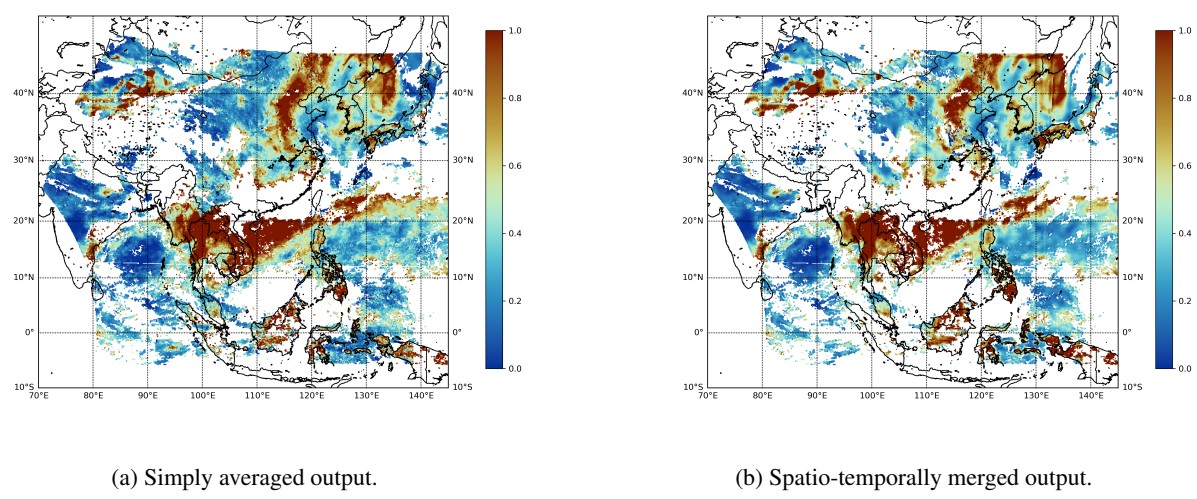

(a) Simply averaged output.          (b) Spatio-temporally merged output.

**Figure B1.** Daily mean-field estimates of GEMS AOD L3 product on April 1, 2023, at the wavelength of 354nm.





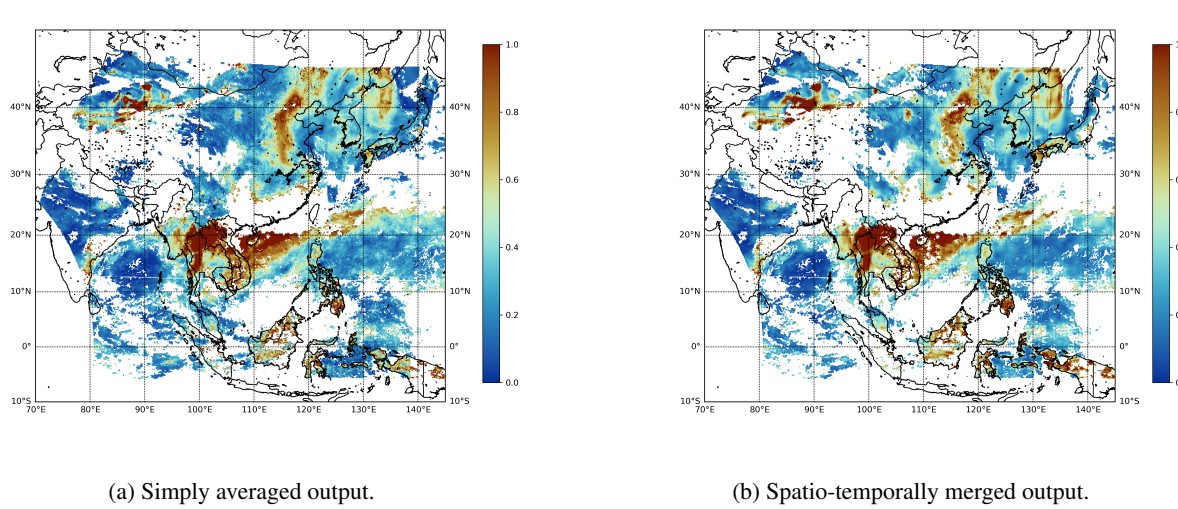

(a) Simply averaged output.

(b) Spatio-temporally merged output.

**Figure B2.** Daily mean-field estimates of GEMS AOD L3 product on April 1, 2023, at the wavelength of 443nm.

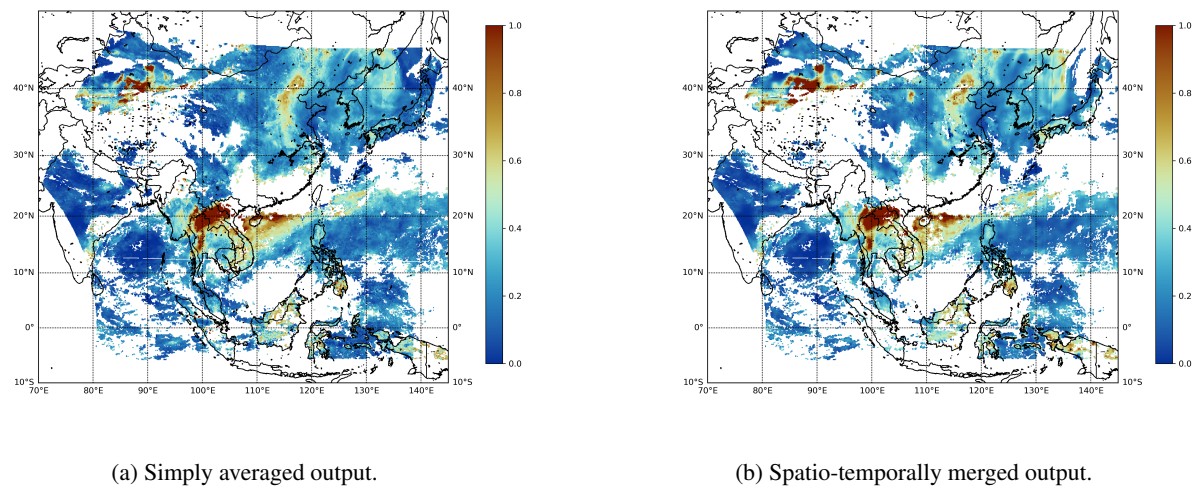

(a) Simply averaged output.

(b) Spatio-temporally merged output.

**Figure B3.** Daily mean-field estimates of GEMS AOD L3 product on April 1, 2023, at the wavelength of 550nm.



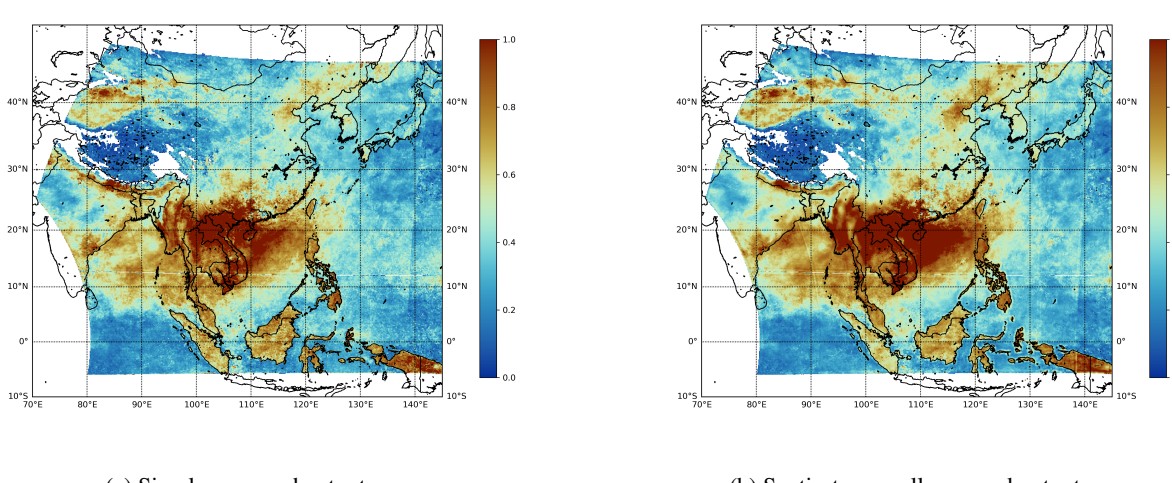

(a) Simply averaged output.

(b) Spatio-temporally merged output.

**Figure B4.** Monthly mean-field estimates of GEMS AOD L3 product in April, 2023, at the wavelength of 354nm.

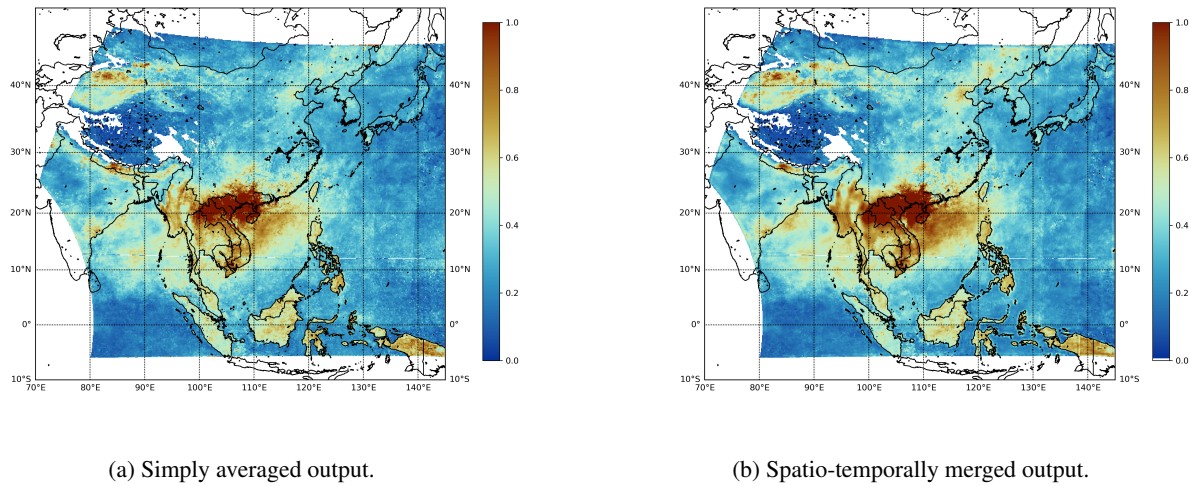

(a) Simply averaged output.

(b) Spatio-temporally merged output.

**Figure B5.** Monthly mean-field estimates of GEMS AOD L3 product in April, 2023, at the wavelength of 443nm.



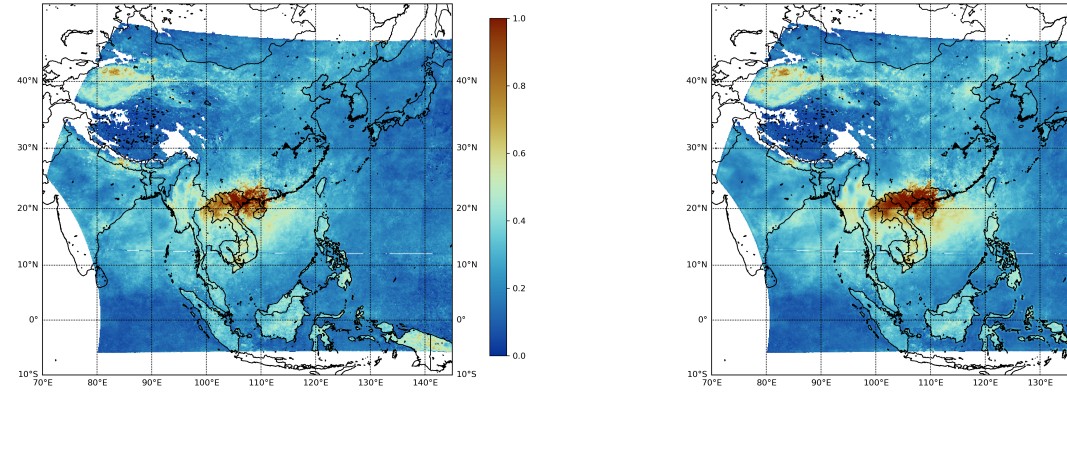

(a) Simply averaged output.          (b) Spatio-temporally merged output.

**Figure B6.** Monthly mean-field estimates of GEMS AOD L3 product in April, 2023, at the wavelength of 550nm.




## Appendix C: Mean Field Estimates of GEMS AOD L3 Product for Korea

In this section, we include daily and monthly mean-field estimates of GEMS AOD L3 products for three wavelengths for Korea.

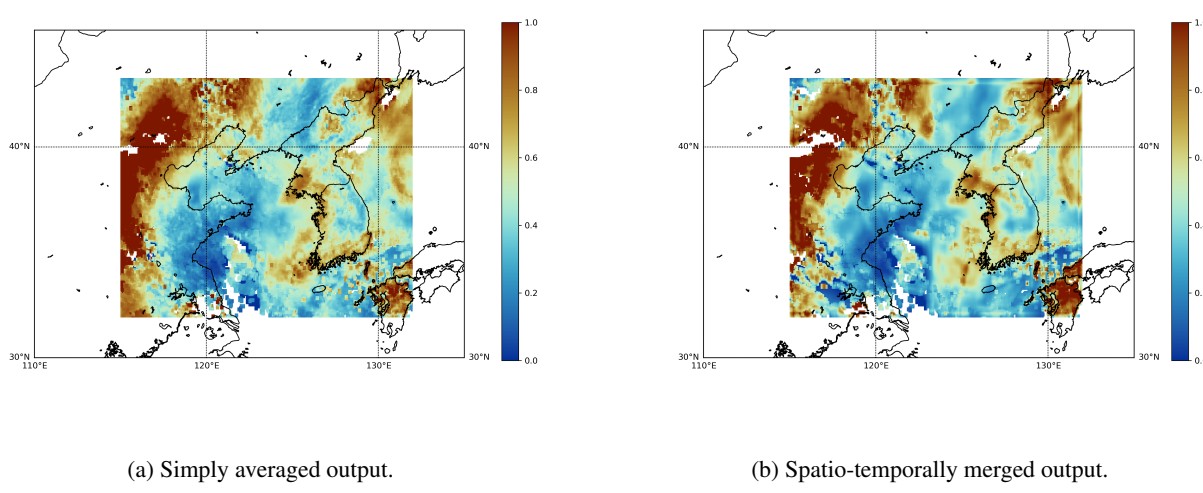

(a) Simply averaged output.

(b) Spatio-temporally merged output.

**Figure C1.** Daily mean-field estimates of GEMS AOD L3 product on April 1, 2023, at the wavelength of 354nm.

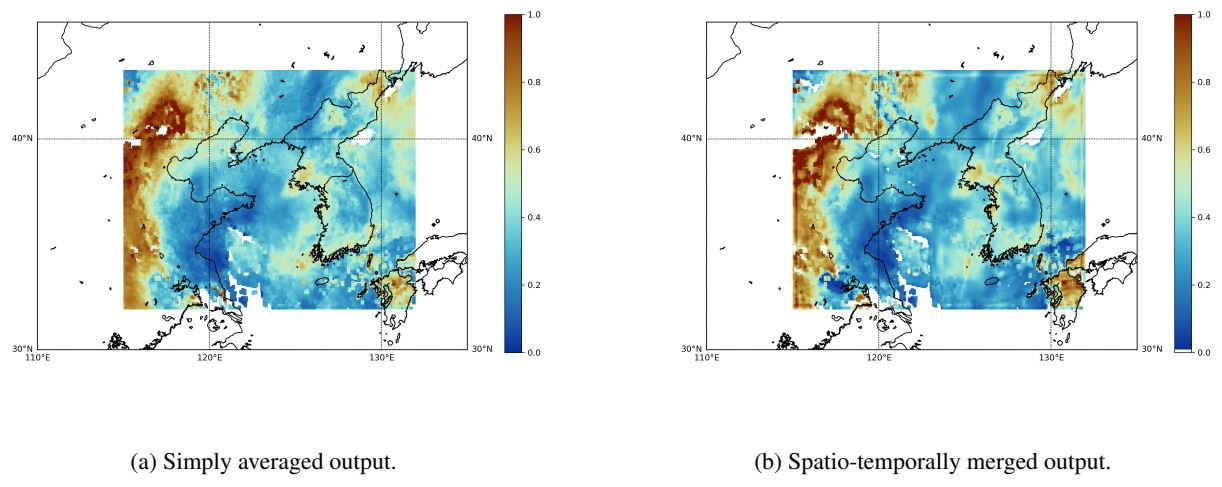

(a) Simply averaged output.

(b) Spatio-temporally merged output.

**Figure C2.** Daily mean-field estimates of GEMS AOD L3 product on April 1, 2023, at the wavelength of 443nm.



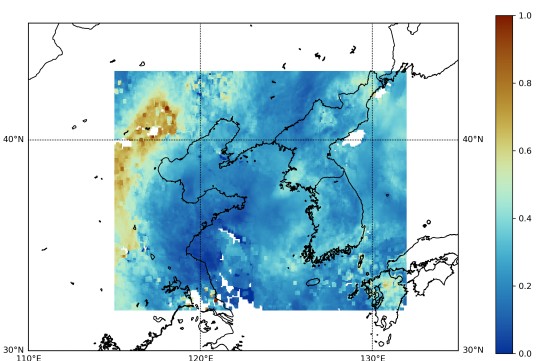
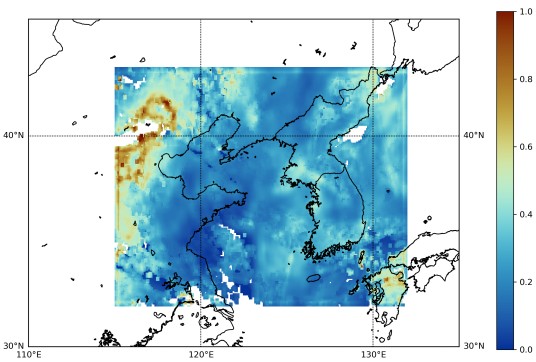

(a) Simply averaged output.  (b) Spatio-temporally merged output.

**Figure C3.** Daily mean-field estimates of GEMS AOD L3 product on April 1, 2023, at the wavelength of 550nm.

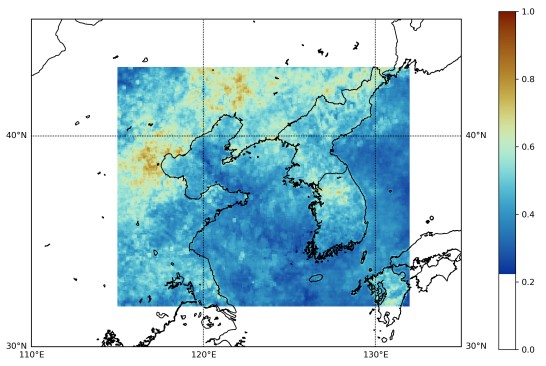

(a) Simply averaged output.  (b) Spatio-temporally merged output.

**Figure C4.** Monthly mean-field estimates of GEMS AOD L3 product in April, 2023, at the wavelength of 354nm.





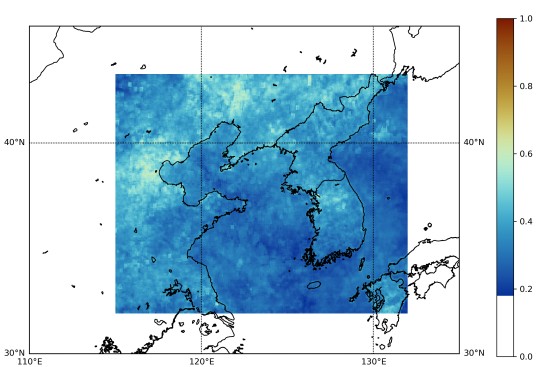
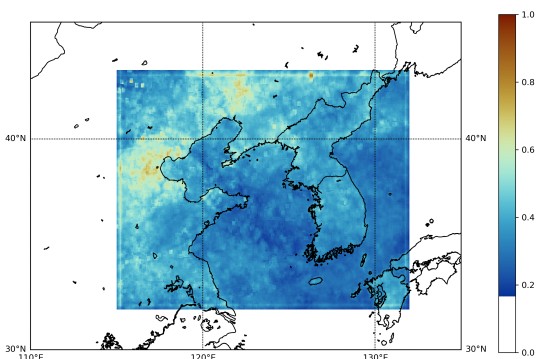

(a) Simply averaged output.                    (b) Spatio-temporally merged output.

**Figure C5.** Monthly mean-field estimates of GEMS AOD L3 product in April, 2023, at the wavelength of 443nm.

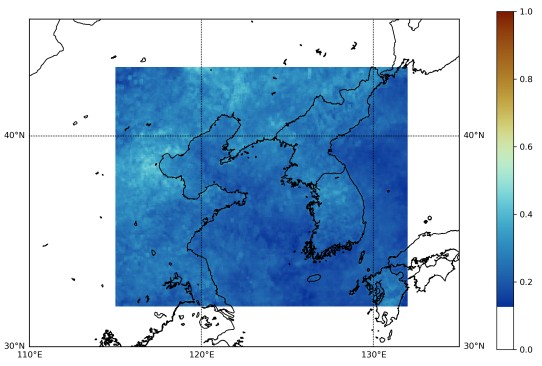

(a) Simply averaged output.                    (b) Spatio-temporally merged output.

**Figure C6.** Monthly mean-field estimates of GEMS AOD L3 product in April, 2023, at the wavelength of 550nm.



*Author contributions.* GEMS data were provided by YSC, JK and WK. All authors participated in developing the overall algorithm design based on a conceptual idea of a Spatio-temporal merging algorithm by SK. Specifically, Spatio-temporal merging algorithms were developed by SK and SP with the guidance of JP and IHJ. Simulation studies of quality flags incorporated in Spatio-temporal merging algorithms were conducted by SL and DO. Data analyses were made by YHC and HK. The paper was written, edited, and proofread by all the authors.

*Competing interests.* The contact author has declared that none of the authors has any competing interests.

*Acknowledgements.* This work was supported by the National Research Foundation of Korea (2020R1C1C1A0100386814, RS-2023-00217705, RS-2023-00218377), ICAN (ICT Challenge and Advanced Network of HRD) support program (RS-2023-00259934) supervised by the IITP (Institute for Information & Communications Technology Planning & Evaluation). The authors are grateful to the anonymous reviewers for their careful reading and valuable comments.



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
