# Peer review of "Improved Mean Field Estimates of GEMS AOD L3 Product: Using Spatio-temporal Variability"

_EGUsphere, 2024_

## Author Comment (AC1)

**Author Responses to Manuscript ID: egusphere-2024-604**

**Improved Mean Field Estimates of GEMS AOD L3 Product: Using Spatio-temporal Variability**

We are grateful to the reviewers for their insightful comments and thought-providing questions. In response to their feedback, we have prepared a comprehensive point-by-point explanation. Furthermore, we have made significant revision efforts to our manuscript, incorporating their suggestion to improve the overall quality and clarity of our revised manuscript. We trust that the editors and reviewers will find the revised manuscript to be significantly enhanced and more effectively conveys the significance of our research.

**1 Responses to Referee's Comments : RC1**

1. In Section 3.1, the radius is set to be 0.1°, but the actual area covered by a circle with a 0.1° radius varies significantly depending on the latitude bands. Since the GEMS AOD data analyzed here cover a large spatial domain, this can introduce some unexpected artifacts. I am wondering if the authors have considered such issues.

   **Response**: We appreciate the reviewer's attention to these details. In our preliminary study, we found that the output is not sensitive to the choice of radius, as our analysis primarily focuses on East Asia region. Figure 1 compares between the outputs obtained from 0.1° radius and a 10km radius. The spatial patterns exhibited by both outputs are highly similar, indicating that the choice of radius does not significantly impact the overall results within the scope of our study.

2. In Section 3.2.2, I do not quite follow the motivation and the description of the regression-based method. If I understand correctly, the method is regressing the computed average variability $\sigma_{\mathrm{IDW}}$ on different radius values. If so, why is it stated that "the spatio-temporal variability in Equation (4) becomes small as the spatial or temporal distance between grids becomes larger"? Perhaps what is more relevant is the fact that average variability is highly correlated with the number of points used to compute the variability. Also, why is the second-order design matrix used rather than other possibilities?

[Figure]

Figure 1: Daily mean-field estimates of GEMS AOD L3 product on April 1, 2023 with 0.1° radius (left) and 10km radius (right).

**Response**: Thank you for pointing this out. It was our mistake. Spatio-temporal variability increases as the distance between grids increases. We have now fixed the typo. The previous

[Figure]

Figure 2: Figure 3 in Kikuchi et al. (2018)

work in Kikuchi et al. (2018) showed that there is a quadratic relationship between the RMSD of AOD estimates and the spatio-temporal distances (Figure 2). In our study, we also follow this convention.

*(Section 3.2.2, instead of "small")* *Spatio-temporal variability increases as the distance between grids increases.*

3. Figure 4 shows that the IDW method clearly leads to oversmoothing. I think the authors need to discuss this issue.

**Response**: In our initial analysis, we set the radius size as 9, which can lead to oversmoothing when applied to the simulated datasets. To address this issue and maintain consistency with the real data application, we have now adjusted the radius size to 3 in the main manuscript. This modification has proven effective in mitigating the oversmoothing problem, ensuring that the simulated results more accurately represent the underlying patterns. To maintain transparency and provide a comprehensive overview, we have relocated the IDW results obtained using a radius size of 9 to the Appendix.

*(Section 4.3) After careful consideration and analysis, we have chosen to set the neighboring order to 3 for our study because varying the order of neighbor did not yield any significant differences in the mean squared error (MSE). However, to provide a comprehensive understanding of the impact of order on the interpolation process, we have included additional IDW results with increasing order sizes in Appendix A. These supplementary results demonstrate that larger order sizes can potentially lead to oversmoothing.*

**References**

Kikuchi, M., H. Murakami, K. Suzuki, T. M. Nagao, and A. Higurashi (2018). Improved hourly estimates of aerosol optical thickness using spatiotemporal variability derived from himawari-8 geostationary satellite. *IEEE Transactions on Geoscience and Remote Sensing 56*(6), 3442–3455.

---

## Author Comment (AC2)

**Author Responses to Manuscript ID: egusphere-2024-604**

**Improved Mean Field Estimates of GEMS AOD L3 Product: Using Spatio-temporal Variability**

We are grateful to the reviewers for their insightful comments and thought-providing questions. In response to their feedback, we have prepared a comprehensive point-by-point explanation. Furthermore, we have made significant revision efforts to our manuscript, incorporating their suggestion to improve the overall quality and clarity of our revised manuscript. We trust that the editors and reviewers will find the revised manuscript to be significantly enhanced and more effectively conveys the significance of our research.

**1 Responses to Referee's Comments : RC2**

**1.1 General Comments**

1. Specifically, this study uses quality flags to weigh observations, not to filter them out. Since this is not a usual approach to employing quality flags, a justification is expected to be described somewhere in the manuscript. In the averaging step of the proposed algorithm, a quality flag representing the presence of clouds is chosen as a weighting factor. However, in the proceeding step, the algorithm filters out pixels with cloud fractions > 0.4 anyway. These two conflicting approaches raise the question of why not simply filter out all pixels with issues.

   **Response**: Thank you for your comment. To improve the performance of the IDW algorithm, we incorporate quality flag information. However, it is still necessary to filter out pixels with unreliable values that can lead significant uncertainties in mean field estimates. To clarify this point, we have added the following explanation to the manuscript.

   *(Section 1) Incorporating quality flag information can reduce the influence of low-quality data on the IDW algorithm, resulting in lower MSE, as we demonstrated in our simulation study. Nevertheless, it remains crucial to identify and exclude pixels with unreliable AOD values, as they can introduce substantial uncertainties in mean field estimates. In our analysis, we apply a filter to remove pixels with a cloud radiance fraction (CRF) exceeding 0.4, ensuring that the included AOD values are not significantly impacted by cloud contamination.*

2. The manuscript says that spatio-temporal variability is crucial to consider but does not explic-itly explain why. One possible reason could be that aerosols are supposed to vary smoothly in time and space, so abrupt variations can be regarded as anomalies. For example, some aerosol retrieval algorithms take advantage of that fact to segregate aerosols from clouds. It will benefit readers if some statements are added about why spatio-temporal variations matter in data filtering.

**Response**: Thank you for the valuable comment. Following the reviewer's suggestion, we have added the importance of considering spatio-temporal variability.

*(Section 1) Numerous studies have demonstrated that aerosol optical depth (AOD) exhibits significant spatio-temporal variability due to natural and anthropogenic factors. This is par-ticularly evident in regions like northwestern China, where understanding the spatio-temporal dynamics is essential for effective atmospheric pollution management and control, as aerosol concentrations are heavily influenced by seasonal and graphical variations (Zhang, 2023). Moreover, the characteristics of the data collection device, the satellite, play a crucial role. The AOD data analyzed in this study is obtained from the Geostationary Environment Mon-itoring Spectrometer (GEMS). When collecting data using such devices, it is imperative to consider spatio-temporal variability to ensure the high-quality of aerosol data. For instance, a study conducted in East Asia optimized the spatio-temporal ranges used for validating satellite products, such as total ozone and NO2, by leveraging long-term data from both ground-based and satellite observations (Park et al., 2020).*

3. Lastly, the effectiveness of the proposed method is mostly assessed qualitatively and subjec-tively. Without further highlights, visual inspections of the entire GEMS domain maps do not tell much about the quality improvements compared to the simple averages.

**Response**: We agree with the reviewer that the quantitative evaluation for the mean field products would be necessary. To address this, we calculate the second-order gradients of the AOD estimates from longitude and latitude. Specifically, we compute gradients at each pixel point from its neighboring AOD values. These gradients can measure the smoothness of the AOD surfaces. In our study, we compare the absolute mean of gradients from the simple and spatio-temporal averaging methods. We observe that the gradients are smaller in our method, indicating that the spatio-temporal merging method can provide smoother AOD surfaces. Considering that AOD exhibits significant spatio-temporal variability, our method can provide more realistic mean field products. To explain this, we have now added Section 5.5 as follows.

*(Section 5.5) To measure the smoothness of mean field products, we compute the second-order gradients of the AOD estimates from longitude and latitude. Specifically, we compute gradients at each pixel point from its neighboring AOD values following previous studies (Fornberg, 1988; Quarteroni et al., 2007; Durran, 1999). Then, we compare the absolute mean of the gradients from the simple and spatio-temporal averaging methods. Table 6 indicates that gradients from both directions are smaller in the spatio-temporally merging method. Considering that AOD exhibits significant spatio-temporal variability, our method can provide more realistic mean field products than the simple averaging method.*

**1.2   Specific Comments**

1. Line 18: GEMS is a satellite instrument (or sensor, payload), not the satellite itself. The satellite is GK-2B, as properly described by the authors in the later part. Therefore, I suggest a slight revision of the expression.

   **Response**: You're correct in pointing out that GEMS is not the satellite itself but rather a satellite instrument or payload. The satellite hosting the GEMS instrument is GEO-KOMPSAT-2B (GK-2B). As stated in the later part of the text, GEMS was launched in February 2020 onboard the GEO-KOMPSAT-2B satellite. We revised the expression over the article.

2. Lines 43–44: How do you define "realistic"? I suggest using more objective terms here to describe the strength of the proposed algorithm.

   **Response**: As we pointed out in the previous response, our method provides smoother AOD surfaces by considering spatio-temporal variability (Section 5.5). We have now edited the sentence as follows.

   *(Section 1, instead of "realistic") We observe that our method can provide more smoother AOD surfaces than the simple averaging method without considering spatio-temporal variability.*

3. Line 85: The mathematical expressions at the end of this line (the ranges of $x_i$ and $y_i$) imply that the algorithm uses a square, not a circle, in which case a "radius" is not a correct term. Please clarify.

   **Response**: For clarification, we have now redefined $r$ as the distance between the point of interest and its $k$-th order neighbor, and we have revised the expression "radius" throughout the article.

4. Equation (3): I understand that the lack of quantified AOD uncertainties led to using quality

flags as weighting factors. However, this approach raises two major discussion points.

First, quality flags are usually meant to be used for simply "filtering out" data points rather than weighting them. I strongly recommend discussing why the authors chose the weighting method over the filtering method, as well as how the Level-3 data quality would be impacted by choosing the weighting method instead of the filtering method.

Second, this approach assumes that the issue represented by each bit of quality flag quantitatively has an equivalent impact on AOD uncertainty. That's the implication of a simple summation of different quality flag bits (the equation in Line 116). In reality, however, the quantified impacts should be different between different issues. Please discuss this point in the manuscript.

**Response**: Thank you for the insightful comments. Regarding the use of quality flags as weighting factors rather than for filtering, the decision was influenced by the real-time updates of the GEMS product, which compromised the reliability of the quality flags as sole criteria for data exclusion. Additionally, the varying ratio of bad samples across pixels suggested that filtering out all samples indicated by the quality flags could potentially lead to significant data loss. To mitigate this, we employed more conservative criteria and algorithms—such as Cloud Radiance Fraction (CRF), Solar Zenith Angle (SZA), Viewing Zenith Angle (VZA), and a proposed spatio-temporal merging algorithm—to exclude unreliable pixels. Thus, quality flags were used as auxiliary information to better leverage the available data without disproportionately discarding useful observations.

Your second point raises a crucial aspect of our methodology. You correctly pointed out that our approach assumes all issues represented by the quality flag bits have an equivalent impact on the uncertainty of AOD estimates, as implied by the summation of different quality flag bits in Line 116 in the original manuscript. We acknowledge that this assumption simplifies the complexities inherent in the data quality issues and may not accurately reflect their individual impacts on the AOD uncertainty. Addressing this limitation would require a differentiated assessment of each quality flag bit's impact, a significant methodological enhancement we intend to pursue in future work. This future research will aim to integrate other data sources and develop a more nuanced understanding of each flag's quantitative impact on the overall data quality. This discussion has been added to the manuscript to clarify our approach and outline the scope for further detailed analysis.

*(Section 6) ⋯ Additionally, with the available information on quality flags, one direction for future work is to develop methodologies for adaptively weighting or selecting quality flag bits*

*by employing statistical variable selection methodologies.*

5. Lines 126–127: Why is it crucial to consider spatio-temporal variability? Although the authors provided a reference (Kikuch et al., 2018), I strongly recommend describing it in this manuscript as well, at least briefly. Readers can easily have the following question: What happens if the variability is not accounted for?

   The Introduction Section also says that the spatio-temporal variability should be considered, without saying why. Given that the term "spatio-temporal" is even included in the manuscript title, its importance should be described somewhere. The Introduction might be one of the proper sections for it.

   **Response**: Thank you for your comment. We have added the motivation of considering spatio-temporal variability. Please see our response to Reviewer 2, General Comments 2 in page 4.

6. Equation (6): The inverse of the variability is used for the weighting factor. This approach assumes that the "variability" represents the "uncertainty," as described in Lines 146 and 147. This approach can be justified because high variability is often due to cloud contamination.

   However, what if the variability is real (physical)? For example, let's say there was a wildfire that produced a lot of aerosols. In that case, the aerosol plumes can have large spatial and temporal variabilities, and they can be accurate and reliable. Could it be considered a limitation of the approach? Please discuss this aspect.

   We are grateful for the insightful comment. Due to the limited data sources in our current study, we cannot use additional physical mechanisms in the interpolation step. We have now added the following to Section 6 as future research avenues.

   *(Section 6) Our method can account for variability due to cloud contamination by utilizing quality flag information in the IDW estimates. Note that we cannot use physical mechanisms (e.g., aerosols produced from a wildfire) in the interpolation step due to the limited data sources. Developing extensions of our approach by incorporating physical mechanisms may provide interesting future research avenues.*

7. Line 149: The following sentence in the manuscript could benefit from a backup explanation: "Note that the spatio-temporal variability in Equation (4) becomes small as the spatial or temporal distance between grids becomes larger." Why is that the case?

**Response**: Thank you for highlighting this issue. Spatio-temporal variability becomes larger as the distance between grids becomes larger. We have now fixed the typo.

*(Section 3.2.2, instead of "small") Spatio-temporal variability increases as the distance between grids increases.*

8. Lines 186–187: The acronym RMSE appears without presenting the full name. The full name for the acronym RMSD is presented above in the manuscript, though. Also, the acronym MSE appears many times without the full name, starting from Line 213. Please give the full names, although the meanings of both RMSE and MSE are not hard to guess.

   **Response**: We have changed the manuscript accordingly.

9. Lines 190–191: Shouldn't these opening lines be placed under Section 4, not 4.1? There is no mention of the "choice of quality flags" in this section (4.1).

   **Response**: We have replaced the opening lines under Section 4.

   *(Section 4) In this section, we conduct a simulation study to validate the choice of quality flags. The data generation procedure is summarized as follows.*

10. Line 192 (Step 1): There should be a description of each mathematical term.

    Specifically, I have four questions: (a) What does the matrix $X$ represent? (i.e., what are the elements?) (b) Why the number of elements should be 4900 × 2? (c) What is $U$? and (d) What is $\beta$?

    I recommend naming each variable first in an English expression and then introducing the corresponding mathematical expression. The former is missing for Step 1 in the current manuscript, although the following Steps have them. Regarding question (b), maybe the description of a $70 \times 70$ lattice in Step 2 can be relocated to an earlier part.

    **Response**:

    (a) The matrix $X$ represents the coordinate matrix of the corresponding locations.

    (b) In our simulation setup, we consider a 70×70 lattice with grid unit 0.1, resulting in a total of 4,900 locations. Each location is characterized by two coordinates: longitude and latitude. Consequently, the dimensions of the $X$ matrix are 4,900 (total number of locations) × 2 (longitude and latitude).

(c) The notation $U$ represents the uniform distribution. $X \sim U(0,1)$ indicates that the elements of the $X$ matrix are sampled from uniform distribution over $[0,1]$.

(d) To improve clarity, we have added an English explanation of the simulation data setup before presenting the mathematical expressions. Additionally, we have relocated the description of the 70×70 lattice to the earlier part of the section, as suggested, to provide a clearer context for the reader.

*(Section 4.1)*

*(a) For our simulation study, we constructed a $70 \times 70$ lattice over a $1 \times 1$ square domain, with a grid spacing of $0.1$. Each grid within this lattice represents a location in our simulated dataset. In total, the lattice comprises 4,900 grid cells, covering an area of $7 \times 7$. Consequently, each unit grid on the lattice represents an area of $0.1 \times 0.1$. We generate each element of $\mathbf{X} \in \mathbb{R}^{4900 \times 2}$ from Unif$(0,1)$ (i.e. the uniform distribution with support [0,1]). This means that we have 4900 locations, each containing two pieces of coordinates information: longitude and latitude. We use $\boldsymbol{\beta}_{1 \times 2} = (1,1)$ as a true coefficient vector.*

*(b) For each location, we simulate zero-mean Gaussian process $\boldsymbol{W}$ from $N(0, (\tau \mathbf{M}' \mathbf{Q} \mathbf{M})^{-1})$ where $\mathbf{M}$ is obtained by taking the first $k$ eigenvectors of the Moran operator (Hughes and Haran, 2013) with smoothness parameter $\tau$. Here, $\mathbf{Q} = \text{diag}(\mathbf{A1})$ - $\mathbf{A}$ is a precision matrix calculated from the adjacency matrix, $\mathbf{A}$. Note that $\mathbf{1}$ is all-ones vector.*

*(c) We simulate AOD datasets from $\boldsymbol{Y} = \boldsymbol{X}\boldsymbol{\beta} + \boldsymbol{W} \in \mathbb{R}^{4900 \times 2}$. In our simulation, $\boldsymbol{X}\boldsymbol{\beta}$ represents the fixed effect, while $\boldsymbol{W} \in \mathbb{R}^{4900}$ account for spatial correlation in AOD products.*

*(d) To generate missing data for the simulated $\boldsymbol{Y}$ that resemble the GEMS L2 product, we apply the observed missing pattern from the GEMS AOD data to the simulated $\boldsymbol{Y}$ from Step 3. Specifically, we apply the missing data pattern observed in the GEMS AOD data for the 7° × 7° region in East Asia, with a grid spacing of 0.1°, collected on April 1st at 04:45. This selected dataset aligns with a 70×70 lattice in our simulation and contains approximately 20% missing values, effectively replicating the realistic missing data characteristics found in the actual GEMS AOD observations.*

*(e) For a realistic simulation that incorporates the physical implications of quality flags, we adapt the observed quality flags from the GEMS AOD data to the simulated $\boldsymbol{Y}$ from Step 3. Similar to Step 4, we use the quality flag of the same data (i.e., 7° × 7° (with a unit 0.1°) in East Asia, observed on April 1st at 04:45) for the simulated $\boldsymbol{Y}$.*

11. Line 195 (Step 2): What is A1? Also, what is the unit for the [0, 1] domain? (The second question also applies to (0, 1) and (1, 1) in Step 1.)

   **Response**: As illustrated, **A** represents the adjacency matrix and **1** denotes the all-ones vector. The unit for the support $[0, 1]$ domain is $0.1°$. The explanation for the $(0, 1)$ is addressed above, and $(1, 1)$ is the matrix notation of the vector. An $N$-dimensional vector $\boldsymbol{v}$ can be specified in either of the following forms using matrices as below:

$$\mathbf{v} = [v_1, v_2, ..., v_n] = (v_1, v_2, ..., v_n)$$

   *(Section 4.1) We generate each element of $\mathbf{X} \in \mathbb{R}^{4900 \times 2}$ from Unif$(0, 1)$ (i.e. the uniform distribution with support [0,1]). This means that we have 4900 locations, each containing two pieces of coordinates information: longitude and latitude. We use $\boldsymbol{\beta}_{1 \times 2} = (1, 1)$ as a true coefficient vector.*

12. Line 200 (Step 4): How can you adopt the missing pattern from a geolocated AOD map, when your grid units are not longitude and latitude? In other words, how can readers match the 70 X 70 lattice with the actual GEMS observation domain?

   **Response**: We apologize for any confusion by our previous explanation. In the simulation study, we focus on a specific $7°\times7°$ region in East Asia, using data observed on April 1st at 04:45. Given that the grid spacing is set to $0.1°$, this selected region corresponds to a $70\times70$ lattice. To incorporate realistic missing data patterns into our simulated dataset, we apply the missing value pattern from the selected real-world data to the simulated data. As a result, the simulated dataset contains approximately 20% missing values, mimicking the characteristics of the actual GEMS AOD data. To clarify this point, we have revised the relevant sentence in the manuscript as follows:

   *(Section 4.2) Specifically, we apply the missing data pattern observed in the GEMS AOD data for the $7° \times 7°$ region in East Asia, with a grid spacing of $0.1°$, collected on April 1st at 04:45. This selected dataset aligns with a $70\times70$ lattice in our simulation and contains approximately 20% missing values, effectively replicating the realistic missing data characteristics found in the actual GEMS AOD observations.*

13. Line 213: In which step this MSE value is calculated? Between what variables? Is it between the simulated AODs and the IDW results?

   **Response**: MSE is calculated between the simulated AOD values and the corresponding IDW

results. We have added the definition of MSE. It can be written as below:

$$\frac{1}{n}\sum_{i=1}^{n}(\text{AOD}_i - \text{IDW}_i)^2$$

where $\text{AOD}_i$ is the simulated AOD values on location $i$ and $\text{IDW}_i$ is the IDW result on location $i$.

*(Section 4.3) Here, MSE is calculated between the simulated AOD dataset and IDW results, which is given as*

$$\frac{1}{n}\sum_{i=1}^{n}(AOD_i - IDW_i)^2$$

*where $AOD_i$ is the simulated AOD values on location $i$ and $IDW_i$ is the IDW result on location $i$.*

14. Lines 218–219: If the spatial unit has always been the same within Section 4, this unit description should be placed in Section 4.1 before describing the detailed Steps. It's not easy to fully understand the Steps in Section 4.1 without information on the unit (see also comment 11 above).

    **Response**: Following the reviewer's suggestion, we have now added the explanation related to the spatial unit in Section 4.1.

    *(Section 4.1) For our simulation study, we constructed a $70 \times 70$ lattice over a $1 \times 1$ square domain, with a grid spacing of $0.1$. Each grid within this lattice represents a location in our simulated dataset. In total, the lattice comprises 4,900 grid cells, covering an area of $7 \times 7$. Consequently, each unit grid on the lattice represents an area of $0.1 \times 0.1$.*

15. Figure 4: The algorithm proposed in this study implicitly involves gap-filling (right panel). Although this fact is described in Lines 186 and 290, I suggest adding a sentence somewhere in Lines 37–44 in the Introduction, where the authors describe the proposed method in short. The reason for the suggestion is that gap-filling is not a capability that every Level-3 algorithm has. It would be helpful for readers to be aware of the characteristics at an early stage.

    **Response**: Thank you for the suggestion. We have added the explanation in Section 1.

    *(Section 1) In this study, we employ the IDW algorithm to compute mean field estimates for AOD data. The IDW method is a widely used gap-filling algorithms that imputes missing observations through a linear combination of neighboring observations.*

16. Lines 227–231: I had a hard time following the quality flag experiments. How can simulation results have quality flags? If the authors simulated them along with AOD values, the process

should be described in Section 4.1. Also, how realistic the simulated quality flags can be? As presented in Table 2, each bit has a physical implication. Do the simulated quality flags reflect the physical meaning?

**Response**: As the reviewer pointed out, simulating realistic quality flags is very challenging because they are related to the missing patterns. To address this issue, we have incorporated the observed missing pattern and quality flag from the actual GEMS AOD data into our simulated experiment, as illustrated in Figure 2. Specifically, we have selected 7°×7° region over East Asia, using data observed on April 1st at 04:45. We then applied the missing pattern and quality flags in the selected data to the simulated data. For further details, please refer to our response to Reviewer 2, Q12. To clarify this methodology, we have included the following explanation in Section 4.1 of the revised manuscript.

*(Section 4.1, instead of "Same as above..") Similar to Step 4, we use the quality flag of the same data (i.e., 7°×7° (with a unit 0.1°) of the East Asia, using data observed on April 1st at 04:45), for the simulated $Y$.*

17. Figure 5: What does the y-axis represent? I suppose it's MSE, but please consider presenting it in the figure. Also, what do the whiskers and horizontal bars represent? Please describe the definitions in the caption.

    **Response**: In Figure 5, the y-axis represents MSE values obtained from the simulation study. To visualize the distribution of MSE values, we employ a box plot, a widely used statistical tool for summarizing and comparing datasets. The box itself is drawn from the first quartile (Q1) to the third quartile (Q3) of the MSE values, with a horizontal line inside the box denoting the median value across the repeated simulations. The whiskers extend from the box to the most extreme data points that are not considered outliers. In this case, the lower whisker ends at the minimum observed MSE value, while the upper whisker extends to the maximum observed MSE value.

    *(Section 4.3) Boxplot with interquartile range describing the IDW algorithm results on 100 repetitive simulation datasets, including bit 0 to bit 15 respectively in uncertainty metric $\sigma(i)$. The boxes represent the interquartile range since they are drawn from Q1 to Q3, with a horizontal line drawn inside to denote the median of the repeated simulations. The boundary of the lower whisker is the minimum value of the MSE. In contrast, the boundary of the upper whisker is the maximum value of the MSE among repeated simulations. We discover that bit 2 and bit 6 have lower MSE compared to other bits.*

18. Line 235: I suppose the implication here is that the presence of clouds and out-of-range SSA/AOD affect the Level 3 quality the most. To just echo comment 4, please discuss the advantage of keeping those pixels without filtering them out.

    **Response**: To emphasize the advantage of keeping the pixels, we have added further explanation as mentioned in the general comment # 1.

    *(Section 1) Incorporating quality flag information can reduce the influence of low-quality data on the IDW algorithm, resulting in lower MSE, as we demonstrated in our simulation study. Nevertheless, it remains crucial to identify and exclude pixels with unreliable AOD values, as they can introduce substantial uncertainties in mean field estimates. In our analysis, we apply a filter to remove pixels with a cloud radiance fraction (CRF) exceeding 0.4, ensuring that the included AOD values are not significantly impacted by cloud contamination.*

19. Line 242: The pixel size of 7 km $\times$ 8 km is not relevant to this study. As stated in Line 52, the GEMS pixel size is 3.5 km $\times$ 7.7 km. Also, the first sentence of Section 5.1 fits more as a concluding sentence. Please consider simply moving the position of the opening sentence to the last part of the paragraph. Otherwise, the description of how you should choose the grid size makes a reader expect to see a different conclusion (e.g., possible changes in the grid size).

    **Response**: As you pointed out, 7 km $\times$ 8 km is not relevant to our study. It was to say that we set the mean field spatial resolution small, but we found it unnecessary and deleted. Also, we moved the first sentence to a concluding sentence according to your suggestion.

    *(Section 5.1) Accordingly, in our study, we set the mean field spatial resolution as 0.1° $\times$ 0.1° and 0.05° $\times$ 0.05° longitude-latitude grid for the East Asia and Korean Peninsula region, respectively.*

20. Line 243: I recommend putting "sensor" or "instrument" after "A geostationary satellite" in this line (this comment aligns with comment 1).

    **Response**: We have changed the manuscript accordingly.

21. Line 253: It says that the algorithm masks data based on quality flags, but this section (5.2) does not give any information on what quality flags were used for that purpose. The criteria were given only for three physical variables, not quality flags. Also, the reference in line 254 (Choi et al., 2020) is for the cloud product, not the aerosol product. From which product does the algorithm extract the quality flags? Please consider elaborating on this matter.

Furthermore, this section (5.2) does not explicitly mention the use of the GEMS L2 cloud product. Most of the information on this use is given in Lines 63–68. Also, this section (5.2) says that the cloud fraction was used as a filtering parameter, but Lines 63–68 say that the algorithm uses the cloud "radiance" fraction. Please consider reorganizing and clarification.

**Response**: Two incorrect representations were corrected—one where something was mistakenly referred to as 'quality flags' instead of 'variables', and another where something was mistakenly referred to as 'cloud fraction' instead of 'cloud radiance fraction'. And the masking process we're explaining in Section 5.2 was based on the three 'variables': cloud radiance fraction, solar zenith angle and viewing zenith angle. We also newly mentioned that cloud radiance fraction is from GEMS L2 cloud product and the others are from GEMS L2 aerosol product, which is explained more in Section 2.

*(Section 5.2) To address these complexities, this study involved masking data based on three variables given below. Cloud radiance fraction is incorporated within GEMS L2 cloud product, and the others are incorporated within GEMS AOD L2 product.*

22. Line 257: Is this consideration (i.e., filtering out unreliable data) applied to the simulation experiments presented in Section 4?

Bit 6 in the aerosol quality flag (the presence of clouds) and the threshold here (cloud fraction $geq$ 0.4) have consistent implications. Why did the authors choose to filter out not the former but the latter?

Also, given possible overlaps between the two criteria, I imagine that the quality flag bit 6 doesn't play a significant role. A discussion is needed on this point.

**Response**: Although bit 6 in the aerosol quality flag provides a binary indication of cloud contamination, we opted to use the cloud radiance fraction from the GEMS L2 cloud product as our masking criteria. The cloud radiance fraction is a continuous variable that offers a more nuanced representation of the extent of cloud contamination in each pixel.

23. Line 266: The sentence "This procedure [. . . ] smoother output" seems to fit better as the concluding sentence of the section. Would you consider moving this sentence to a later part (after presenting the detailed interpretation first)?

**Response**: We have changed the manuscript accordingly.

24. Section 5.4: This section hardly gives an evaluation of the results. Of course, the challenge can be justified as described in the manuscript. However, the manuscript says that the results are

consistent with "springtime global distribution" and "trends from MODIS" without presenting any actual comparisons.

I'm not suggesting that the manuscript should present the comparisons with independent observations. Those comparisons would be more about the performance of the Level-2 aerosol retrieval algorithm, which is out of the scope of this study.

The main focus of this study is to involve quality flags and spatio-temporal variability for Level 3 processing. Therefore, I suppose the evaluation this study should present is a more detailed comparison with the simple averaging method, not with independent observations (e.g., MODIS). Although some of these comparisons are presented in Section 5.3, they are not sufficient to demonstrate the effectiveness of the proposed method.

I strongly recommend adding more detailed comparisons between panels (a) and (b) in Figs. 7 and 8. Maybe Sections 5.3 and 5.4 can be merged in this step.

Specifically, the following questions can be answered: How many percentages of missing values were recovered compared to the simple average? How can you show quantitatively that the proposed method provides smoother output? Basically, the arguments stated in Section 5.3 are not clearly backed up by only Figs. 7 and 8.

**Response**: Following your suggestion, we have included more detailed comparisons between the spatio-temporally merged products and those obtained through simple average method. Specifically, we have added a new metric, the ratio of missing values, for each figure and discussed it in our manuscript. This ratio is calculated as the number of grids with missing values divided by the number of total grids. Interestingly, we observe that the spatio-temporally merged products have a higher proportion of missing values compared to the simply averaged products. As explained in Section 5.3, this difference arises because the spatio-temporally merging procedure only considers reliable AOD estimates, while the simple averaging method does not discriminate based on reliability. Consequently, we conclude that the simply averaged products may be regarded as less reliable despite having fewer missing values. Furthermore, in the newly added Section 5.5, we provide a quantitative analysis demonstrating that the spatio-temporal merging method yields smoother output compared to simple averaging. This smoothness is indicative of a more consistent and coherent representation of the underlying spatio-temporal patterns in the data, as elaborated upon in our response to General Comment 3.

25. Line 289: By any chance, did you mean to say "levels" instead of "units" in the first sentence?

**Response**: We have changed the manuscript accordingly.

**1.3 Technical Corrections**

1. Line 68: Missing period at the end of the paragraph.

2. Line 94: yields $\rightarrow$ yield

3. Lines 145, 149, 205, 211: Equation $\rightarrow$ Eq.

4. Line 146: Section 3.2.2 $\rightarrow$ Section 3.2.1

5. Line 192: an each element $\rightarrow$ each element

6. Line 282: 550nm $\rightarrow$ 550 nm (with a space)

7. Line 286: Please provide the full name of MODIS.

8. Line 296: ADO $\rightarrow$ AOD

**Response**: Thank you for your attention to details. We have changed the manuscript accordingly.

**References**

Durran, D. R. (1999). *Numerical methods for wave equations in geophysical fluid dynamics / Dale R. Durran.* Texts in applied mathematics. Springer.

Fornberg, B. (1988). Generation of finite difference formulas on arbitrarily spaced grids. *Mathematics of Computation 51*(184), 699–706.

Hughes, J. and M. Haran (2013). Dimension reduction and alleviation of confounding for spatial generalized linear mixed models. *Journal of the Royal Statistical Society Series B: Statistical Methodology 75*(1), 139–159.

Park, S. S., S.-W. Kim, C.-K. Song, J.-U. Park, and K.-H. Bae (2020). Spatio-temporal variability of aerosol optical depth, total ozone and no2 over east asia: Strategy for the validation to the gems scientific products. *Remote Sensing 12*(14), 2256.

Quarteroni, A., R. Sacco, and F. Saleri (2007, 01). *Numerical Mathematics*, Volume 37.

Zhang, F. (2023). Factors influencing the spatio–temporal variability of aerosol optical depth over the arid region of northwest china. *Atmosphere 15*(1), 54.